# Ligand-based design of [18F]OXD-2314 for PET imaging in non-Alzheimer's disease tauopathies

Anton Lindberg[1], Emily Murrell [1,2], Junchao Tong [1], N. Scott Mason[3], Daniel Sohn[4], Johan Sandell[5], Peter Ström[5], Jeffrey S. Stehouwer [3], Brian J. Lopresti[3], Jenny Viklund[4], Samuel Svensson[4] ✉, Chester A. Mathis [3] ✉ & Neil Vasdev [1,2] ✉

Positron emission tomography (PET) imaging of tau aggregation in Alzheimer's disease (AD) is helping to map and quantify the in vivo progression of AD pathology. To date, no high-affinity tau-PET radiopharmaceutical has been optimized for imaging non-AD tauopathies. Here we show the properties of analogues of a first-in-class 4R-tau lead, [18F]OXD-2115, using ligand-based design. Over 150 analogues of OXD-2115 were synthesized and screened in post-mortem brain tissue for tau affinity against [3H]OXD-2115, and in silico models were used to predict brain uptake. [18F]OXD-2314 was identified as a selective, high-affinity non-AD tau PET radiotracer with favorable brain uptake, dosimetry, and radiometabolite profiles in rats and non-human primate and is being translated for first-in-human PET studies.

Tauopathies are a heterogeneous class of neurodegenerative diseases including Alzheimer's disease (AD)[1], chronic traumatic encephalopathy (CTE)[2], progressive supranuclear palsy (PSP)[3], corticobasal degeneration (CBD)[4], and Pick's disease (PiD)[5], characterized by the formation of aggregated filaments of microtubule associated protein tau in the brain. Human tau has six isoforms that are mainly differentiated by having either 3- or 4-repeated microtubule binding regions (3R- or 4R-tau)[6,7]. Tauopathies are distinguished by the relative abundance of 3R- and 4R-tau isoforms in their respective filaments: AD and CTE are mixed 3R/4R tau tauopathies; PSP and CBD are primarily 4R-tauopathies; and PiD is primarily a 3R-tauopathy[8–10]. Recent cryogenic-electron microscopy (cryo-EM) structural studies of tau filaments have revealed different folding patterns that further distinguish the tauopathies[11–15].

The first successful, selective positron emission tomography (PET) radiotracer for imaging amyloid-β (Aβ) plaques was [11C]PiB[16,17], and subsequently led to three PET radiopharmaceuticals approved by the United States Food and Drug Administration (FDA) for imaging this

target in AD: [18F]Amyvid, [18F]Neuroceq, and [18F]Vizamyl[18–20]. The development of PET radiotracers for imaging aggregated tau in AD has recently resulted in [18F]Tauvid™ (a.k.a. [18F]flortaucipir, [18F]AV-1451, and [18F]T807) being the first and only PET radiopharmaceutical to be FDA-approved (2020) for imaging tau deposits in cognitively impaired subjects undergoing clinical evaluation for AD (Fig. 1A)[21]. A next-generation tau PET radiopharmaceutical, [18F]MK-6240 (a.k.a. [18F]florquinitau), has also been shown to effectively image tau aggregates in AD patients (Fig. 1B)[22,23]. PET imaging of tau aggregation in non-AD tauopathies has been limited to utilizing radiopharmaceuticals originally developed and optimized for imaging AD tau aggregates. For example, [18F]PI-2620 (Fig. 1A), a structural analogue of [18F]Tauvid, has been evaluated for human PET imaging studies in non-AD tauopathies and has shown the ability to image tau aggregation in patients with suspected PSP and corticobasal syndrome (CBS), a common clinical expression of CBD, but does not provide highly robust PET imaging outcome measures[24–26]. A PET study of 148 PSP patients using [18F]florzolotau (a.k.a. [18F]PM-PBB3 or [18F]APN-1607, Fig. 1C) has shown

[1]Azrieli Centre for Neuro-Radiochemistry, Brain Health Imaging Centre, Campbell Family Mental Health Research Institute, Centre for Addiction and Mental Health, Toronto, Canada. [2]Department of Psychiatry, University of Toronto, Toronto, Canada. [3]Department of Radiology, University of Pittsburgh, Pittsburgh, PA, USA. [4]Oxiant Discovery, SE-15136 Södertälje, Sweden. [5]Novandi Chemistry AB, SE-15136 Södertälje, Sweden.
✉e-mail: samuel.svensson@oxiantdiscovery.com; mathis@pitt.edu; neil.vasdev@utoronto.ca

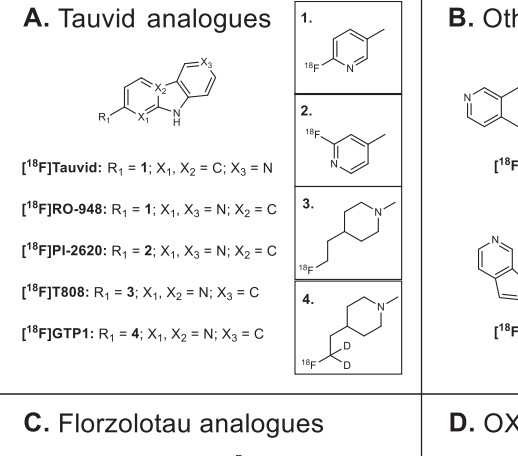

**Fig. 1 | Structures of tau PET radiotracers divided into structural scaffolds.**
**A** Compounds based on the carbazole scaffold of [18F]Tauvid™ (a.k.a., [18F]flortau-cipir or [18F]T807) which also include [18F]PI-2620. **B** Janssen's naphthalene-pyridine linked scaffolds, as well as [18F]MK-6240 ([18F]florquinitau) and [18F]FDDNP. **C** [11C] PBB3, [18F]F-PBB3, and [18F]florzolotau ([18F]PM-PBB3, [18F]APN-1607) analogues based on a benzothiazole scaffold. **D** PET radiotracers based on Oxiant Discovery's fluoropyridinyl-indole scaffold (OXD analogues) including [18F]OXD-2115 (a.k.a. CBD-2115) and [18F]OXD-2314 (this work).

promise to detect tau accumulation typical of PSP pathophysiology; however, longitudinal follow-up studies are necessary to determine whether [18F]florzolotau can be a reliable diagnostic tool for PSP[27]. [18F] Florzolotau also showed uptake and regional distribution in brain consistent with expected tau aggregation in CBS patients[28]. A recent meta-analysis of [18F]florzolotau-PET imaging in patients with PSP revealed the need for PET radiopharmaceuticals with higher selectivity for 4R-tau over 3R-tau and other CNS targets, including α-synuclein and Aβ, in order to detect disease-specific distribution of tau aggregates[29].

Our laboratories recently developed a first-in-class PET radiotracer optimized for 4R-tau, [18F]OXD-2115 (a.k.a. [18F]CBD-2115; Fig. 1D)[30,31]. Unfortunately, [18F]OXD-2115 showed low brain uptake in rodents and non-human primate (NHP) in vivo. However, [3H]OXD-2115 has been shown to be a useful radiotracer for 4R-tau aggregates in PSP tissues in vitro[32]. Additionally, OXD-2115 was computationally shown to have favorable binding characteristics for CTE tau protofibrils[33]. OXD-2115 represents a new structural scaffold for 4R-tau PET radiotracer development with high selectivity for 4R-tau aggregates over Aβ plaques, α-synuclein deposits, and other CNS targets.

In the present study, pyridinyl-indole analogues of OXD-2115 were identified using ligand-based design to select for high potency and selectivity for non-AD tau aggregates as well as improved brain uptake. Over 150 analogues were synthesized for initial screening for tau affinity. Lead candidates were further evaluated against [3H]OXD-2115 in human postmortem AD and PSP tissues for comparison with first- and second-generation PET radiopharmaceuticals for imaging tau aggregation (Table 1). Computational prediction tools were used to assess the candidates' ability to cross the blood-brain barrier (BBB). OXD-2314 (2-(2-fluoro-6-(3-methoxypiperidin-1-yl)pyridin-3-yl)-1H-indol-5-ol) was identified as the most promising compound and was radiolabeled with 3H for in vitro assays in AD, CBD, PSP, PiD, Parkinson's disease (PD) and healthy control brain tissues (patent application public in May 2024).

Further in vitro pharmacological evaluations to determine off-target binding to CNS receptors and enzymes were undertaken. [18F]OXD-2314 was radiolabeled and evaluated by PET imaging in rats and NHPs to assess brain uptake and clearance. Radiometabolite and dosimetry studies were carried out in preparation for human imaging in non-AD tauopathies.

## Results

### Synthesis and identification of pyridinyl-indole analogues

Synthesis of structural analogs of OXD-2115 focused on diversification of either the $R_1$ substituent on the indole hydroxyl group or the $R_2$ substituent on the central pyridinyl ring[34]. With the aim to reduce the number of hydrogen bond donors (HBDs), adding alkyl groups on the indole hydroxyl moiety were explored. Derivatization of the aliphatic ring on in the $R_2$ position was explored to reduce HBDs as well as modify the lipophilicity. The lipophilicity of the molecule was modified by adding nitrogen in either one or both of the aromatic rings. A library of over 150 compounds were synthesized based on OXD-2115 and screened against [3H]OXD-2115 in AD and PSP tissues. Compounds with $K_i$ values below 200 nM were selected for further screening. Two analogues of OXD-2115, the enantiomeric pair, OXD-2188 and OXD-2189, were initially identified as promising candidates. [18F]OXD-2188 and [18F]OXD-2189 were synthesized and both showed low brain uptake in rat PET imaging studies and were not evaluated further (Fig. S1). Subsequently, six new leads (OXD-2310, OXD-2312, OXD-2313, OXD-2314, OXD-2317 and OXD-2331) were identified which inhibited [3H]OXD-2115 binding in PSP tissue more efficiently than self-inhibition using unlabeled OXD-2115 whilst also being equally or more potent in AD tissue (Table 1).

### BBB permeability

There are several computational approaches for obtaining high BBB permeability of central nervous system (CNS) PET radiotracers[35]. In

**Table 1 | Inhibition constant ($K_i$) values ($n = 1$) of OXD-2115 and analogues vs [³H]OXD-2115 in AD and PSP tissue homogenates to establish inhibition potencies**

| Compound | AD tissue $K_i$ (nM) | PSP tissue $K_i$ (nM) | ClogP | CNS MPO | CNS PET MPO | BBB score |
|---|---|---|---|---|---|---|
| OXD-2115 | 20 | 27 | 2.35 | 3.7 | 1.9 | 3.18 |
| OXD-2310 | 21 | 6.6 | 1.74 | 4.2 | 3.0 | 3.16 |
| OXD-2331 | 7.4 | 6.3 | 2.48 | 4.1 | 1.9 | 3.17 |
| OXD-2317 | 5.4 | 5.8 | 3.52 | 3.7 | 1.8 | 4.07 |
| OXD-2312 | 12 | 5.1 | 3.66 | 3.6 | 1.7 | 4.07 |
| OXD-2313 | 11 | 4.8 | 3.66 | 3.6 | 1.7 | 4.07 |
| OXD-2314 | 8.8 | 4.8 | 3.25 | 3.8 | 1.7 | 3.65 |

Chemdraw Professional 15.0 was used for all physicochemical properties except logD$_{7.4}$, which was obtained experimentally.
In silico models for predicting BBB permeability included CNS MPO, CNS PET MPO and BBB scores.

addition to the preliminary assays, in silico calculations were made to predict the BBB permeability and lipophilicity of each lead compound. ClogP was first used to rank the leads based on lipophilicity. The use of computational in silico models offers a fast and high throughput screening method for identifying physicochemical contributors that might affect the BBB permeability of a radiotracer[36]. Three computational models were employed to evaluate potential improvements to the low brain uptake of OXD-2115: CNS MPO, CNS PET MPO, and BBB score[37–39]. OXD-2115 scored 3.7/6.0 in CNS MPO, 1.9/6.0 in CNS PET MPO, and 3.18/6.0 in BBB score, which are all within the range for potentially CNS active compounds in each model (Fig. S2). It is noteworthy that the values acquired when using these models can differ depending on the software or experimental methods used to acquire

the physicochemical values and did not use the same software as the models were trained with. In this study, ChemOffice™ was used for all physicochemical properties except logD$_{7.4}$, which was measured experimentally. Evaluating the subscores for physicochemical properties of OXD-2115, the number of HBDs, the pKa of the indole nitrogen[40], and molecular size are likely physiochemical contributors to the low brain uptake seen in PET scans using [¹⁸F]OXD-2115[30]. CNS MPO and CNS PET MPO also predicted that reducing the lipophilicity would increase the likelihood of a successful CNS drug/PET radiotracer, respectively. However, two radiotracers were explored with lower lipophilicities, [¹⁸F]OXD-2188 and [¹⁸F]OXD-2189, and neither showed improved brain uptake compared with [¹⁸F]OXD-2115 in rats. The BBB score was therefore considered for identifying the next lead.

**Fig. 2 | Synthetic route to OXD-2314 and [¹⁸F]OXD-2314.** Top: chemical precursors **3** or **7** for **OXD-2314** and **[¹⁸F]OXD-2314**, respectively, were synthesized from 2,6-difluoropyridine or 2-chloro-6-nitropyridine according to the route described in the top panel under the following reaction conditions: (a) Hünig's base, dioxane, 100 °C, 5 h (75% yield); (b) N-bromosuccinimide, acetonitrile, 0 °C, 30 min (96% yield); (c) [1,1′-Bis(diphenylphosphino)ferrocene]dichloropalladium(II) (Pd(dppf)Cl₂), potassium carbonate (K₂CO₃), dioxane, 90 °C, 1 h (98% yield); (d) tetrabutylammonium fluoride, tetrahydrofuran, 0 °C, 10 min (67% yield). Each step was followed by flash column purification to isolate each intermediate compound. Bottom: Synthesis of OXD-2314 and radiochemical synthesis of [¹⁸F]OXD-2314 from **3** or **7**, respectively, under the following reaction conditions: (e) methanol, 150 °C, 45 min under microwave heating (47% yield); (f) cyclotron-generated [¹⁸F]F⁻, K₂CO₃, Kryptofix₂.₂.₂, dimethyl sulfoxide, 160 °C, 20 min; followed by addition of methanol, 130 °C, 20 min. [¹⁸F]OXD-2314 was isolated by semi-preparative high-performance liquid chromatography and formulated in 10% ethanol in sterile saline with radiochemical purities of 95–98%. Toronto: Molar activities of $61.5 \pm 26.3$ GBq/µmol ($n = 4$), and radiochemical yields of $8.7 \pm 3.7\%$ (not decay-corrected, ndc); Pittsburgh: molar activities of $349 \pm 144$ GBq/µmol ($n = 4$) and radiochemical yields of $2.3 \pm 1.1\%$ (ndc).

Primarily due to reducing the number of HBDs, four leads showed improved BBB scores compared to OXD-2115 (OXD-2317, OXD-2312, OXD-2313 and OXD-2314; see Table 1) and were equipotent in AD and PSP tissue. However, OXD-2317, OXD-2312 and OXD-2313 had higher lipophilicity (clogP), which may lead to higher non-specific binding and/or plasma protein binding in vivo. OXD-2314 was identified as the most promising high-affinity compound for PSP-tau with potential of BBB permeability and was carried forward for further evaluation (Table 1).

## Chemistry
OXD-2314 was designed and synthesized in 5-steps starting from 2,5-difluoropyridine (Fig. 2). The first step is a nucleophilic aromatic substitution reaction with (S)-3-methoxypiperidine to form 2-fluoro-6-[(3 S)-3-methoxypiperidin-1-yl]pyridine (**1**) in 75% yield. Compound **1** was brominated using N-bromosuccinimide (NBS) to give 3-bromo-2-fluoro-6-[(3 S)-3-methoxypiperidin-1-yl]pyridine (**2**) in 96% yield followed by a Suzuki coupling reaction with 1-[(t-butoxy)carbonyl]-5-[(t-butyldimethylsilyl)oxy]-1H-indol-2-yl boronic acid yielding t-butyl 5-[(t-butyldimethylsilyl)oxy]-2-{2-fluoro-6-[(3 S)-3-methoxypiperidin-1-yl]pyridin-3-yl}-1H-indole-1-carboxylate (**3**) in 98% yield. Subsequent deprotection of the tert-butyl dimethylsilane and t-Boc groups in two steps gave OXD-2314 in 39% yield over the 2 steps.

The precursor to [¹⁸F]OXD-2314, tert-butyl 5-hydroxy-2-{6-[(3 S)-3-methoxypiperidin-1-yl]-2-nitropyridin-3-yl}-1H-indole-1-carboxylate (**7**), was synthesized in an analogous way to OXD-2314 in 4-steps starting from 2-chloro-6-nitropyridine in 26% yield (Fig. 2).

## In vitro binding assays
Scatchard analysis of [³H]OXD-2314 binding in human brain tissue homogenates provided equilibrium dissociation constant ($K_d$) values, which were compared to [³H]PI-2620, [³H]florzolotau, and [³H]OXD-2115 (Table 2). [³H]OXD-2314 bound to 4R-tau aggregates in PSP tissues with a $K_d$ of $2.4 \pm 0.6$ nM and in CBD tissues with a $K_d$ of $2.1 \pm 0.3$ nM. [³H]OXD-2314 was also found to bind non-selectively to 3 R/4R-tau aggregates in AD tissues with a $K_d$ of $3.6 \pm 0.7$ nM and to 3R-tau aggregates in PiD tissues with a $K_d$ of $1.1 \pm 0.2$ nM. [³H]OXD-2314 showed lower binding in PD tissue ($62 \pm 8$ nM) and no specific binding could be quantified in healthy control tissue. [³H]OXD-2314 had higher affinity to tau aggregates than [³H]PI-2620, [³H]florzolotau, and [³H]OXD-2115 in PSP, CBD, and AD tissues and has >10-fold higher affinity in CBD tissue. The sole exception being that [³H]PI-2620 had higher affinity to tau aggregates in AD tissue. Overall, [³H]OXD-2314 was shown to be a high affinity ligand for tau aggregates in both AD and non-AD tauopathies. It is noteworthy that [³H]OXD-2115 showed high and similar affinities to the primarily 4R-tau (PSP, CBD) aggregates in vitro but had lower affinity to the predominantly 3R-tau aggregates in PiD, and [³H]OXD-2314 was expected to be similar, but instead showed high affinity for all of the non-AD and mixed 3 R/4R-tauopathies (AD) in human brain homogenate affinity assays (Table 2).

The $B_{max}$ values of [³H]OXD-2314 in AD, PSP, CBD, PiD, and PD frontal cortex tissue were measured (Table 3) and resulted in the highest $B_{max}$ value in AD tissue at 1990 nM, while PSP, CBD, and PiD frontal cortex $B_{max}$ values ranged from 1390 to 780 nM. PD frontal cortex provided a lower $B_{max}$ value of 240 nM. Subcortical (putamen and globus pallidus) PSP and CBD $B_{max}$ values were higher than cortical values at 2300 and 1600 nM, respectively, while the $K_d$ values of cortical and subcortical PSP and CBD tissues were nearly identical (Table 2). Representative homologous binding assay results of [³H]OXD-2314 in AD, PiD, PSP, CBD, and control brain tissues reveals two-site binding (high and low affinity) in all brain tissues except control (Fig. S24).

Additional binding assays were conducted in AD tissue homogenates to assess the affinity of unlabeled OXD-2314 in competition

**Table 2 | In vitro equilibrium dissociation constant ($K_d$) values ($n = 3$; mean ± SD) of [³H]PI-2620, [³H]florzolotau, [³H]OXD-2115, and [³H]OXD-2314 in AD, PSP, CBD, PiD, and PD frontal cortex tissues as well as P301L transgenic mouse tissue**

| Radiotracer | AD $K_d$ (nM) | PSP $K_d$ (nM) | CBD $K_d$ (nM) | PiD $K_d$ (nM) | PD $K_d$ (nM) | P301L transgenic mice $K_d$ (nM) | Control $K_d$ (nM) |
|---|---|---|---|---|---|---|---|
| [³H]PI-2620 | 2.5 ± 1.8 | 5.3 ± 4.6 | 23 ± 4 | n.d. | n.d. | n.d. | Undefined |
| [³H]Florzolotau | 4.1 ± 0.5 | 4.6 ± 1.0 | 36 ± 7 | 20 ± 3 | 5.1 ± 1.1 | n.d. | Undefined |
| [³H]OXD-2115 | 5.5 ± 1.9 | 4.9 ± 0.2 | 27 ± 1 | 34 ± 12 | 48 ± 11 | n.d. | Undefined |
| [³H]OXD-2314 | 3.6 ± 0.7 | 2.4 ± 0.6 2.3 ± 0.3* | 2.1 ± 0.3 2.2 ± 0.4* | 1.1 ± 0.2 | 62 ± 8 | 5.1 ± 0.4 | Undefined |

See Table S1 for demographics and neuropathology of subjects. n.d. not determined, undefined insufficient specific binding to fit the data. The AD, PSP, CBD, and PiD homogenates were pooled from three cases, the PD homogenates were pooled from 4 cases, and the Control was a representative single subject. *Subcortical tissue.

**Table 3 | $B_{max}$ values for [³H]OXD-2314 in AD, PSP, PiD and PD tissues**

| Radiotracer | Brain region | AD ($n = 3$) $B_{max}$ (nM) | PSP ($n = 3$) $B_{max}$ (nM) | CBD ($n = 3$) $B_{max}$ (nM) | PiD ($n = 3$) $B_{max}$ (nM) | PD ($n = 1$) $B_{max}$ (nM) |
|---|---|---|---|---|---|---|
| [³H]OXD-2314 | Frontal cortex | 1990 ± 890 | 1390 ± 680 | 780 ± 440 | 980 ± 110 | 240 |
| | Subcortical | - | 2300 ± 600 | 1600 ± 100 | | |

**Table 4 | Inhibition constant ($K_i$) values ($n = 1$) of unlabeled OXD-2115, PI-2620, florzolotau, Tauvid, MK-6240, and OXD-2314 against [³H]OXD-2314 and unlabeled OXD-2314 against [³H]OXD-2115 in AD, PSP, and CBD tissue**

| | Competitor | AD $K_i$ (nM) | PSP $K_i$ (nM) | CBD $K_i$ (nM) | PiD $K_i$ (nM) |
|---|---|---|---|---|---|
| [³H]OXD-2314 | OXD-2115 | 14 | 27 | 30 | 23 |
| | PI-2620 | 460 | 620 | 660 | 760 |
| | Florzolotau | >10,000 | 4200 | >10,000 | >10,000 |
| | Tauvid | >10,000 | >10,000 | >10,000 | 4400 |
| | MK-6240 | 1900 | 5600 | 1700 | 1000 |
| | OXD-2314 | 4.8 ± 0.3[a] | 5.8 ± 0.2[a] | 6.5 ± 0.4[a] | 5.5 ± 0.5[a] |
| [³H]OXD-2115 | OXD-2314 | 8.8 | 4.8 | 4.4 | - |
| [³H]PiB | OXD-2314 | 161 ± 34[a] | - | - | - |

[a]$n = 3$.

with [³H]PiB binding to aggregated Aβ (Table 4). The $K_i$ of OXD-2314 vs. [³H]PiB in AD tissue was 161 ± 34 nM ($n = 3$, mean ± SD), indicating a relatively poor affinity of OXD-2314 for the PiB binding site on aggregated Aβ. This also indicates that the high affinity binding of [³H]OXD-2314 to AD tissue shown in Table 2 was dominated largely by binding to 3 R/4R-tau aggregates and not to Aβ plaques. Competition binding assays of [³H]OXD-2314 against unlabeled OXD-2115, PI-2620, florzolotau, Tauvid, MK-6240, OXD-2115 as well as unlabeled OXD-2314 in AD, PSP, CBD and PiD tissue were carried out to establish their inhibition constant ($K_i$) values (Table 4). [³H]OXD-2314 was effectively displaced by OXD-2115 in AD tissue with a $K_i$ value of 14 nM, in PSP tissue with $K_i$ value of 27 nM, in CBD tissue with $K_i$ value of 30 nM and in PiD tissue with $K_i$ value of 23 nM. However, PI-2620, florzolotau, Tauvid and MK-6240 were unable to effectively displace [³H]OXD-2314 in any of the tissues evaluated. OXD-2314 displaced [³H]OXD-2314 in AD tissue with a $K_i$ value of 4.8 ± 0.3 nM, in PSP tissue with $K_i$ value of 5.8 ± 0.2 nM, in CBD tissue with $K_i$ value of 6.5 ± 0.4 nM and in PiD tissue with $K_i$ value of 5.5 ± 0.5 nM. The lack of displacement of [³H]OXD-2314 binding to tau filaments by PI-2620, florzolotau, Tauvid and MK-6240 in AD, PSP, CBD and PiD tissues suggests that OXD-2314 binds to a different high-affinity site on tau aggregates that is not specific to a particular isoform of tau or specific to a particular pathological expression of tau.

Displacement of [³H]OXD-2314 binding with OXD-2115 in AD, PSP and CBD tissues was expected due to their structural similarity, although the higher potency of [³H]OXD-2314 compared to [³H]OXD-

2115 was evident when comparing the $K_i$ values of OXD-2314 competed against [³H]OXD-2115 to OXD-2115 when competed against [³H]OXD-2314.

Autoradiography studies were conducted with [³H]OXD-2314 in healthy control, AD, CBD and PSP tissues for total radiotracer signal and blocking with unlabeled OXD-2314, and compared to AT8 immunohistochemistry for tau staining (Fig. 3). Signal was displaceable under homologous blocking conditions to the level of non-specific binding present in healthy control. Low specific binding was observed in aged healthy control and hippocampal PSP tissues, whereas elevated radiotracer signal and good specific binding were observed in hippocampal AD as well as cortical PSP and CBD tissues. Radiotracer signals in HC, AD and CBD tissues aligned with IHC staining for phospho-tau. High-resolution autoradiography with [³H]OXD-2314 using nuclear emulsion aligned with staining for phospho-tau deposits demonstrated by IHC in cortical CBD tissue (Fig. 3; S21–S23 and Table S5).

## Pharmacology and pharmacokinetics

Ex vivo evaluation of the BBB permeability of OXD-2314 was performed by measuring the brain-to-plasma ratios in mice. Compound concentrations in plasma (0.03 h, 0.17 h, 0.5 h, 1 h, and 2 h; $n = 3$ per timepoint) were compared with concentrations in brain (0.03 h, 0.17 h, 1 h, and 2 h; $n = 3$ per timepoint) for both OXD-2115 and OXD-2314. The total brain-to-plasma concentration ratios were 1.7 for OXD-2314 and 0.1 for OXD-2115. The free concentration ratio for brain-to-plasma was 0.6 for OXD-2314 and 0.002 for OXD-2115 (Figs. S3–S8 and Tables S2, S3).

The affinity of OXD-2314 towards CNS receptors ($n = 70$) and enzymes ($n = 24$) was tested in pharmacology binding assays at a concentration of 10 μM (Eurofins, France, Figs. S9, S10). OXD-2314 binding to CNS receptors was calculated as a percentage of inhibition of a receptor-specific radioligand. Enzyme inhibition was calculated as the percent inhibition of baseline enzyme activity where inhibition above 50% was considered significant. Of note, OXD-2314 was found to inhibit the binding of the antagonist radioligand (protriptyline) to the norepinephrine transporter(h) by 62.9% and the enzyme activity of 5-lipoxygenase by 50.1% and warrants further investigation to establish $IC_{50}$ values. Norepinephrine transports are expressed in low concentration throughout the brain and its relative concentration are not affected by aging or neurodegeneration[41]. 5-lipoxygenase may be overexpressed in aging brain and play some role in neurodegeneration, which could be necessary to further investigate in vivo[42]. Off-target binding to monoamine oxidase A and B (MAO-A and MAO-B) has been seen with some tau PET radiotracers[43,44]. OXD-2314 showed no

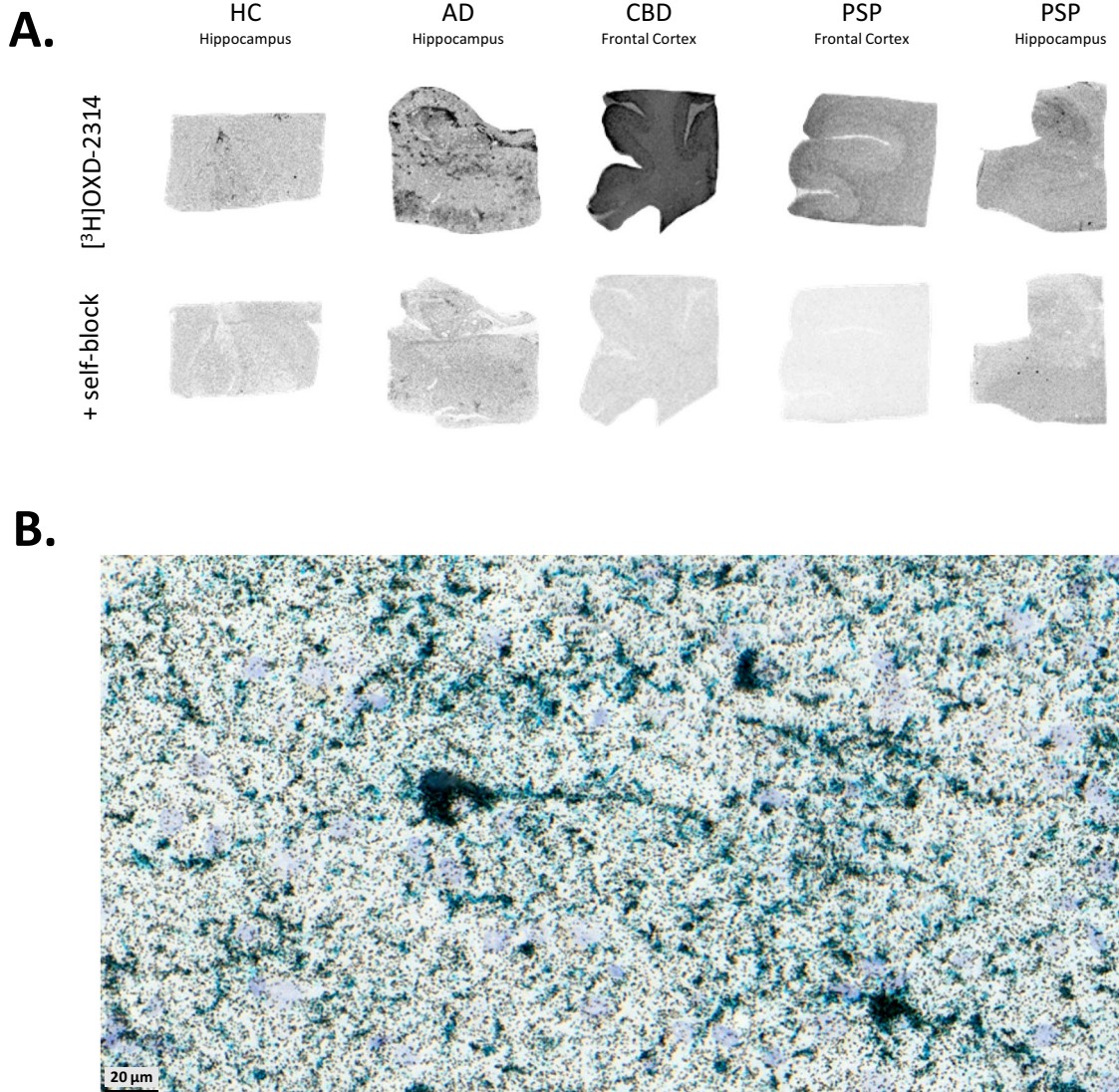

**Fig. 3 | [³H]OXD-2314 autoradiographic binding in human post-mortem brain tissues of tauopathies. A** Autoradiographic detection of [³H]OXD-2314 binding in post-mortem cortical or hippocampal brain tissue samples from healthy control (HC), Alzheimer's disease (AD), corticobasal degeneration (CBD) and progressive supranuclear palsy (PSP) cases. Top row shows total [³H]OXD-2314 (3 nM) binding and bottom row shows non-specific binding of [³H]OXD-2314 (3 nM) with displacement by unlabeled OXD-2314 (3–10 μM) in adjacent sections. Representative data of 2–3 cases per diagnosis. **B** High-resolution autoradiography image of [³H]OXD-2314 (3 nM, black) in CBD cortical tissue. [³H]OXD-2314 signal is overlaid with immunohistochemical (AT8) staining to indicate phospho-tau (turquoise) distribution. Scale bar, 20 μm.

significant interaction with MAO-A (-1.7%) and was found to inhibit binding MAO-B by 37%, which is below the threshold for significance in the binding assay and is considered to be unlikely to cause any pharmacological effects or confounding issues during PET imaging. This assessment is supported by the lack of high affinity binding by [³H]OXD-2314 in healthy control tissue and lack of competitive blocking of [³H]OXD-2314 by either deprenyl or Ro41-1049 (Table 2 and S4).

### PET radiochemistry and imaging

[¹⁸F]OXD-2314 was radiolabeled by an analogous method previously reported for [¹⁸F]OXD-2115 with minor modifications (Fig. 2)[30]. [¹⁸F]OXD-2314 was reliably produced with high radiochemical purity (RCP) ranging from 95-98%, molar activity ($A_m$) of 61.5 ± 26.3 GBq/mol ($n = 4$), and radiochemical yield (RCY) of 8.7 ± 3.7% (not decay-corrected, ndc) in the laboratories in Toronto and was replicated with $A_m$ of 349 ± 144 GBq/μmol ($n = 4$) and RCY of 2.3 ± 1.1% (ndc) in the laboratories in Pittsburgh. The logD$_{7.4}$ value of [¹⁸F]OXD-2314 was measured as 3.11 ± 0.04 using the shake-flask method[45].

Eight rats were used in preclinical PET scans with [¹⁸F]OXD-2314: two for dynamic brain imaging, and six for whole-body biodistribution of radioactivity and dose estimation. Another five rats were used for radiometabolite analysis in brain homogenates and blood plasma. Dynamic PET-CT imaging in healthy wild-type (wt) rats were performed following intravenous administration of [¹⁸F]OXD-2314 (17.9 and 23.8 MBq, $n = 2$, one female and one male). Radioactivity reached an initial peak of 2.3 standardized uptake value (SUV) within two minutes in whole brain, followed by a steady washout of radioactivity in the brain to below 0.3 SUV at the end of the 120 min acquisition time (Fig. 4). Distribution of radioactivity was homogenous among brain regions, but with the female rat having slower clearance of radioactivity from the brain and thus higher overall brain radioactivity than the male rat.

Radiometabolite analysis was performed ex vivo in plasma and brain homogenates from rats ($n = 5$, 3 female and 2 male) at 30 and 60 min after radiotracer injection (Figs. S11–S16). Four polar radiometabolites (M1-M4) were detected in plasma in female rats, whereas

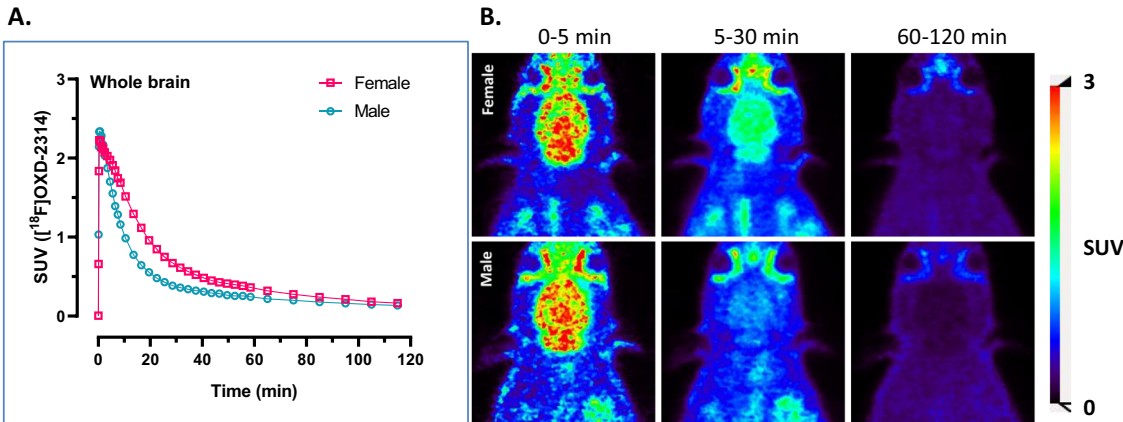

**Fig. 4 | PET imaging of [¹⁸F]OXD-2314 in Sprague-Dawley rats. A** Whole brain time-activity curves of standardized uptake values (SUV) in female (pink, $n=1$) and male (blue, $n=1$) rat whole brains in baseline PET-CT scans using [¹⁸F]OXD-2314 (0–120 min). **B** Representative PET summation images of rat brain from baseline PET scans using [¹⁸F]OXD-2314. Top row shows transverse images of a male wild-type rat brain at 0–5, 5–30, and 60–120 min; bottom row shows transverse images of a female wild-type rat brain at the same time points.

in male rats, only three polar radiometabolites (M1-M3) were observed in blood plasma. The parent fraction in plasma was 6% at 30 min and 5% at 60 min in the two male rats, respectively, and 32% at 30 min ($n=1$) and 18% and 17% at 60 min ($n=2$) in the female rats. Different rates of metabolism between male and female rats may have contributed to the differences in kinetics observed in the brain PET scans. No lipophilic radiometabolites were detected in plasma or brain. Parent [¹⁸F]OXD-2314 accounted for $82\pm7\%$ ($n=2$) and $80\pm5\%$ ($n=3$) of the radioactivity in rat brain homogenates by 30 and 60 min post-injection of the radiotracer, respectively. The rats were not perfused with saline immediately prior to sacrifice, so that radiometabolites in blood were trapped in the brain and likely accounted for a portion of the polar radiometabolites found in brain, as cerebral blood volume in rats is approximately 4–5% of total brain volume[46].

Six rats (3 male, $19.9\pm6.2$ MBq; and 3 female, $20.8\pm3.2$ MBq) were imaged using PET-CT for whole-body biodistribution of radioactivity and dose estimation using standard mass and density of the organs for adult male and female subjects, respectively (Figure S17). The highest uptake of radioactivity was detected in the liver, followed by the small and large intestines ($0.147\pm0.023$, $0.075\pm0.019$, and $0.067\pm0.032$ MBq$_{Organ}\cdot$hr/MBq$_{Injected}$, respectively), suggesting fast hepatobiliary clearance of the radiotracer (Figure S17). As seen in the dynamic brain PET scans, the pharmacokinetics were slightly more rapid in male rats than in females for whole-body scans. The estimated whole-body effective dose was $21.3\pm3.1$ and $22.5\pm0.9$ μSv/MBq for adult male and female models, respectively, with the highest absorbed dose of $50.6\pm16.9$ μGy/MBq in the large intestine.

Dynamic PET imaging was performed in a rhesus macaque (macaca mulatta) monkey following the injection of [¹⁸F]OXD-2314 at baseline (104.9 MBq, $n=1$) and was repeated in a second [¹⁸F]OXD-2314 study (115.8 MBq, $n=1$) where a displacing dose of unlabeled OXD-2314 (0.5 mg/kg) was administered 10 min after [¹⁸F]OXD-2314 injection to verify the absence of specific binding in NHP brain. Venous blood samples were drawn to characterize the metabolic fate of [¹⁸F]OXD-2314 in the macaque. Dynamic [¹⁸F]OXD-2314 PET imaging was also performed in a male (106.6 MBq, $n=1$) and a female (115.8 MBq, $n=1$) baboon (papio anubis), and venous blood samples were drawn in both baboons to characterize potential effects of sex on radiometabolism. Bilateral regions of interest were defined on 5–8 contiguous axial image planes for the lateral temporal cerebral cortex, cerebellum, striatum, and whole brain, which were used to sample dynamic [¹⁸F]OXD-2314 PET images to determine the time-varying radioactivity concentration expressed in SUV units. [¹⁸F]OXD-2314 readily entered macaque brain (whole brain peak ~2.2 SUV at ~2 min)

and distributed uniformly throughout cortical and subcortical grey matter regions (Fig. 5A). Radioactivity cleared rapidly from the whole brain with a clearance half-time of ~22 min and resulted in a 2:90 min radioactivity concentration ratio of 3.9 (Fig. 5B). No preferential retention of [¹⁸F]OXD-2314 in any brain region was observed, suggesting that signal in NHP brain was dominated by non-displaceable radioactivity (non-specifically bound and free radiotracer). Administration of a pharmacologic dose of unlabeled OXD-2314 (0.5 mg/kg) 10 min after [¹⁸F]OXD-2314 injection (Fig. 5C–E) resulted in a very small transient increase in brain radioactivity concentrations, possibly resulting from an increase in plasma concentrations of free [¹⁸F]OXD-2314 released from peripheral sites. There was no evidence of accelerated brain clearance in any brain region that would suggest a significant component of displaceable (specifically bound) [¹⁸F]OXD-2314 to any site in a normal monkey brain. These data are consistent in showing that non-displaceable binding is the only significant source of [¹⁸F]OXD-2314 signal in normal NHP brain. Radiometabolite analysis of macaque venous blood identified only one, polar radiometabolite, which increased from 37% at 10 min to 77% at 90 min post-injection and was not likely brain penetrant (Figs. S19, S20).

The rate of uptake and clearance of [¹⁸F]OXD-2314 in the brains of a male and a female baboon were compared (Fig. 6A). Initial whole brain uptake was rapid in both baboons, with peak uptake achieved in ~2 min with an SUV of ~2.8 in the male baboon and an SUV of ~1.9 in the female baboon. Rapid clearance of brain radioactivity was then observed in both baboons providing a 2:90 min SUV ratio of ~3.7 in the male baboon and ~3.6 in the female baboon. Radiometabolite analysis of [¹⁸F]OXD-2314 in both baboons (Fig. 6B) demonstrated a similar rate of [¹⁸F]OXD-2314 metabolism over the course of the study. It is not known whether the higher initial brain uptake of [¹⁸F]OXD-2314 observed in a male baboon is due to sex differences, normal physiologic variability, or the additional variability introduced by the effects of anesthesia. No evidence of elevated uptake or retention in the baboon striatum following [¹⁸F]OXD-2314 injection was observed (Fig. S16), a region where high levels of PSP and CBD pathology are expected. Therefore, the normal brain retention patterns of [¹⁸F]OXD-2314 are unlikely to confound the detection of specific signal from PSP and CBD pathology in human subjects.

The brain uptake of [¹⁸F]OXD-2314 in rats and NHP's at early times points was above the desired threshold of 1 SUV, with a ratio between the initial maximum of brain radioactivity compared to late minimum brain radioactivity of ~8 in rats and ~4 in NHP (Figs. 4–6). Early-to-late ratios of this magnitude in the brains of healthy animals lacking tau aggregates indicate suitably high brain penetration and rapid

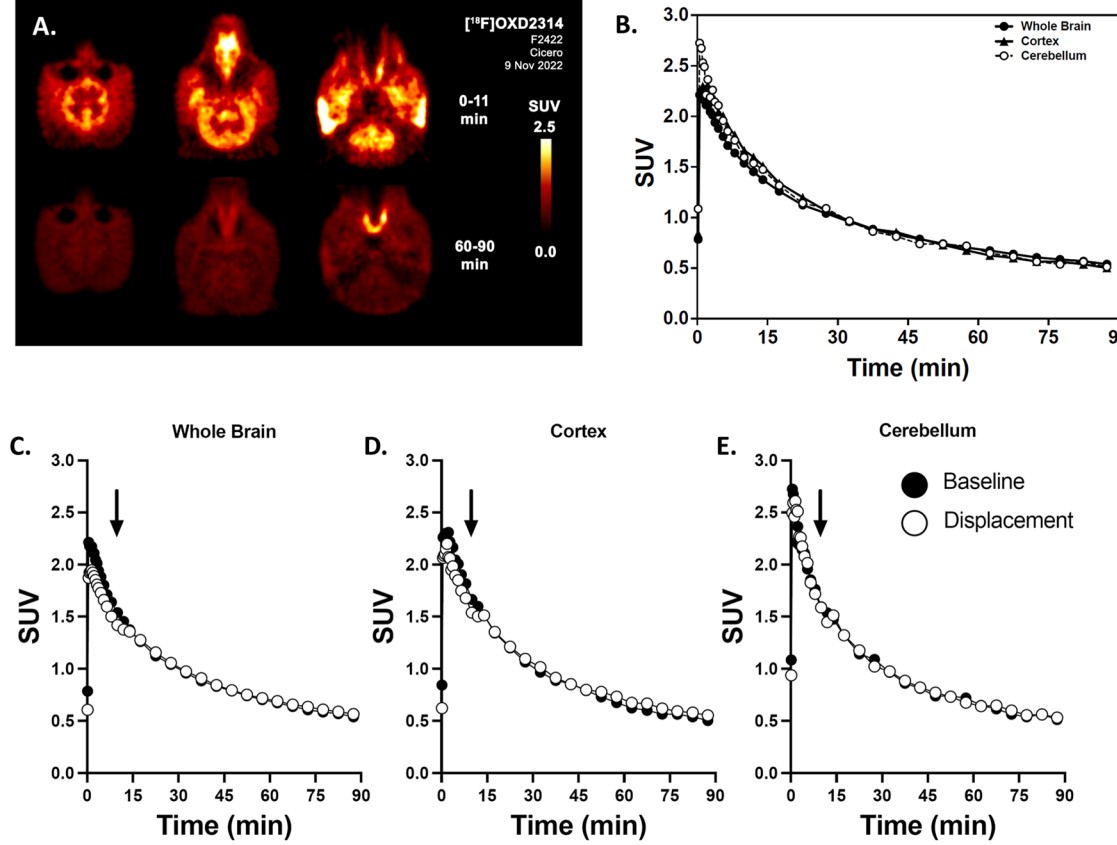

**Fig. 5 | PET imaging of [¹⁸F]OXD-2314 in rhesus macaque monkeys.**
**A** Representative transverse PET summation images (0–11 min and 60–90 min) of macaque brain from baseline PET scan using [¹⁸F]OXD-2314. **B** Time-activity curves of standardized uptake values (SUV) in macaque ($n = 1$) whole brain (solid circles,●), cortex (triangles,▲), and cerebellum (rings, (○) from baseline PET scan using [¹⁸F]OXD-2314 over 0–90 min. **C–E** Time-activity curves of SUV in macaque ($n = 1$) from a displacement PET experiment using [¹⁸F]OXD-2314 with unlabeled OXD-2314, (**C**) whole brain, (**D**) cortex, and (**E**) cerebellum. Baseline (●) and displacement ((○) PET imaging in macaque, 0–90 min. The displacing dose of unlabeled OXD-2314 (0.5 mg/kg) was administered at 10 min post-injection (indicated by arrows). No significant displacement of [¹⁸F]OXD-2314 could be detected in any brain region, as expected in animals without tau pathology.

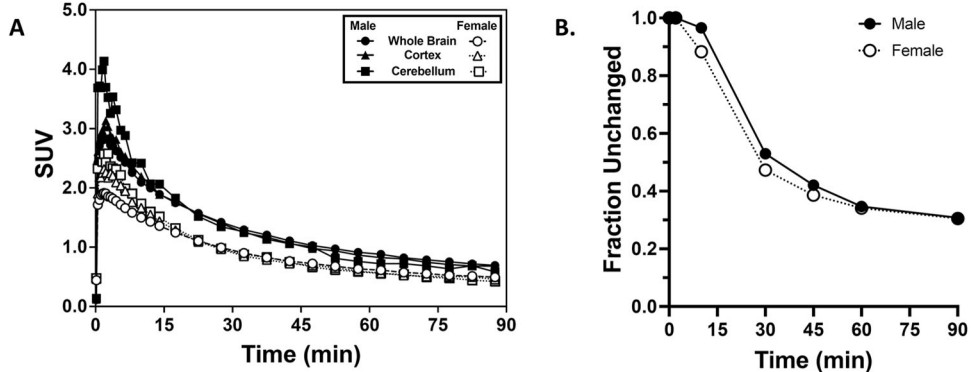

**Fig. 6 | PET imaging and radiometabolite analysis of [¹⁸F]OXD-2314 in baboon.**
**A** Time-activity curves of standardized uptake values (SUV) in male (solids, $n = 1$) and female (open, $n = 1$) baboon whole brain (circles), cortex (triangles), and cerebellum (squares) from baseline PET scans using [¹⁸F]OXD-2314 over 0-90 min. **B** Radiometabolite analysis from male ($n = 1$) and female ($n = 1$) baboon venous blood taken during baseline PET scans using [¹⁸F]OXD-2314. The fraction of unchanged [¹⁸F]OXD-2314 in plasma was measured by high-performance liquid chromatography at six-time points between 2 and 90 min post-injection.

clearance properties desired of an efficacious brain imaging PET radiotracer.

## Discussion

Approximately 150 derivatives of OXD-2115 were synthesized and screened against [³H]OXD-2115 to identify OXD-2314 as a lead candidate for further in vitro evaluation. [³H]OXD-2314 was compared to tritium-labeled PI-2620, florzolotau, and OXD-2115 in affinity and competitive binding assays against AD, CBD, PSP, and PiD tissues and was shown to be twice as potent to tau aggregates in PSP tissue, 10-times more potent in CBD tissue, and equipotent in AD tissues. Surprisingly, [³H]OXD-2314 was also found to have a $K_d$ value of 1.1 nM in PiD tissue indicating potential application in 3R-tauopathies, and in competitive binding assays [³H]OXD-2314 could not be effectively

displaced by either PI-2620 or florzolotau, revealing that this radiotracer binds to a different high affinity site in tauopathies. In vitro pharmacology screening assays showed OXD-2314 had no significant interaction with the 94 CNS receptors and enzymes screened which was further seen by the absence of high affinity binding by [³H]OXD-2314 in healthy control tissues. These results indicate that [³H]OXD-2314 is a high-affinity tau PET radiotracer with unique binding sites on tau aggregates from both AD and non-AD tauopathies.

In silico calculations using the BBB score model indicated that OXD-2314 was likely to be more BBB permeable than OXD-2115. Whereas the CNS MPO and CNS PET MPO scores suggested OXD-2314 and OXD-2115 would be equally suited as CNS drugs/PET radiotracers. Pharmacokinetic studies in mice revealed a higher brain-to-plasma ratio for OXD-2314 compared to OXD-2115. This supports the BBB score model's prediction over those of the CNS MPO and CNS PET MPO score model predictions and is explained by the reduction of HBDs and lower TPSA.

Radiolabeling of [¹⁸F]OXD-2314 was performed at two different laboratories and provided RCY, $A_m$, and RCP results in a range suitable for human PET imaging. Dynamic PET imaging of [¹⁸F]OXD-2314 showed initial brain uptake of >2 SUV in rats and NHP and relatively rapid clearance of radioactivity from the brain as expected for healthy animals with no tau pathology, thereby resulting in an early-to-late brain concentration ratios of ~8 in rats and ~4 in NHP. A limitation of the study is the lack of an appropriate preclinical in vivo model for PET imaging of tau aggregates. In vivo binding to tau with [¹⁸F]OXD-2314 can only be addressed in human PET studies of tauopathies.

Radiometabolite studies in both rat and NHP showed only polar radiometabolites in blood plasma, and ex vivo analysis of brain homogenate from rats showed low radiometabolite levels in the brain samples, indicating that the radiometabolites formed are not readily passing through the BBB. Rat dosimetry studies of [¹⁸F]OXD-2314 indicated rapid hepatobiliary clearance and whole-body doses suitable for translation of [¹⁸F]OXD-2314 to human PET imaging.

In summary, [¹⁸F]OXD-2314 is a brain-penetrant PET radiotracer based on the pyridinyl-indole structural scaffold with high affinity for tau aggregates in AD and non-AD tauopathies. Radiochemistry validation and regulatory submission for translation for first-in-human brain PET imaging studies are underway.

## Methods
### Ethical approvals
**Human tissues.** Postmortem human brain tissues were provided by brain banks at the University of California, San Francisco (UCSF) and Banner/Sun Health AZ following informed consent of the donors and utilized at the University of Pittsburgh under the approval of the Committee for Oversight of Research and Clinical Training Involving Decedents. See Table S1 in Supplemental Information for demographic and neuropathological characterizations of the postmortem tissues.

**Autoradiography and immunohistochemistry (Toronto).** All AD, PSP and healthy control tissue was obtained from the Douglas Bell Canada Brain Bank, in accordance with the guidelines put forth by the Centre for Addiction and Mental Health Research Ethics Board (protocol 036-2019). The sample size was determined by tissue availability.

**PET studies using rats (Toronto).** Rat imaging studies were carried out in accordance with the guidelines put forth by the institutional animal care and use committee at the Centre for Addiction and Mental Health under Animal Usage Protocol #891. Rats are housed in conventional, static caging on paper chip bedding. They are given nesting material, sunflower seeds and shelters for enrichment. The light cycle is 7 am on/ 7 pm off. The humidity is maintained between 30-70%. The room temperature is maintained between 20-25 degrees Celsius. Total room air changes are at a rate of 15-20 times/hour.

**NHP veterinary care, housing, and enrichment program (Pittsburgh).** The University of Pittsburgh is fully accredited by the American Association for Accreditation of Laboratory Animal Care (AAALAC), and the animal husbandry staff at the University of Pittsburgh's Division of Laboratory Animal Resources are devoted to full-time care of animals housed in institutional facilities. In addition, several fully qualified veterinarians are employed by the University of Pittsburgh, and they ensure that the animals receive the best possible care to minimize discomfort, distress, pain, or injury. The male macaque was housed in a cage adjacent to another male macaque so that they could interact as much as possible without harmful physical contact. The male and female baboons were housed together in the same cage to allow social bonding. In the cage environment, species-specific behavior was encouraged by providing foraging feeding methods, perches, food puzzles, and various toys such as such as dental balls and paper for shredding. In addition, diverse food options, including a variety of fresh fruits, vegetables, and treats such as popcorn, peanuts, prima-treats and other items, provided different tastes, smells, colors, sizes, shapes and densities. Visual stimulation was provided by mirrors and a television played in the room during daylight hours. At various times a radio was played to provide additional auditory stimulation. Water was available ad libitum until the time of the PET scans. A full-time environmental enrichment specialist was assigned to the non-human primate floor of the animal facility. Enrichment techniques utilized by the enrichment specialist fell into four category types: treats, foraging, toys, and sensory/human. For example: 1) treats included popcorn, vegetables (carrots, turnips, sweet potato), dried fruit (pineapple, raisins, cranberries), and fresh fruits (grapes, apples, oranges); 2) foraging included puzzles filled with fruit and hay placed on top of the cages; 3) toys stimulated visual and tactile senses and cognitive abilities, and included plastic toys of various shapes and approved destructible paper items; and 4) sensory/ human enrichment included mirrors, television, and radio. Human interaction was provided at least 4 times per week through positive postures, gestures, and spoken interactions, as well as the placing of food treats in the cages. The daily record was monitored by animal caretakers, and the behaviors of all non-human primates were evaluated routinely for signs of psychological distress. Any non-human primate showing signs of psychological distress received special attention in terms of additional environment enrichments from the veterinary staff and the enrichment specialist.

### Experimental Details
**Radiochemistry.** (*Toronto*) No-carrier added [¹⁸F]F⁻ was produced via the ¹⁸O(p,n)¹⁸F reaction (Scanditronix MC-17 cyclotron) and transferred in [¹⁸O]water to an automated radio-synthesizer (GE Tracerlab FX2N) in a lead-shielded hot cell. The [¹⁸F]F⁻ was trapped on a PS-HCO₃ ion exchange column (Chromafix) before being eluted with K₂CO₃ (1.37 mg, 10 μmol) and Kryptofix 2.2.2 (14 mg, 5 μmol) in water/methanol (10/90, 1 mL) into the reaction vessel. Solvents were evaporated under vacuum and subsequently azeotropically dried with acetonitrile (1 mL) under continuous nitrogen flow. The precursor, tert-butyl (*S*)-5-hydroxy-2-(6-(3-methoxypiperidin-1-yl)-2-nitropyridin-3-yl)-1*H*-indole-1-carboxylate (3 mg, 6.4 μmol), in DMSO (1 mL) was added and the reactor was heated to 160 °C for 20 min. The reactor was cooled down to 70 °C and methanol (2 mL) was added before heating to 130 °C for 20 min. The reactor was then cooled to 50 °C and the reaction mixture was diluted with acetonitrile:water (20:80) before being injected onto a reverse-phase HPLC column (LUNA 10 μm C18(2) 100 Å, 250 mm × 10 mm, Phenomenex). The desired product was eluted with a mobile phase of acetonitrile-NH₄CO₂H$_{aq}$ (0.05 M) (45/55, v/v) at a flow rate of 5 mL/min. The retention time of [¹⁸F]OXD-2314 was 14-16 min. The collected fraction was diluted with water (20 mL) and loaded onto a solid phase extraction (SPE) column (SepPak tC18, Waters). The SPE column was washed with water (10 mL) before [¹⁸F]

OXD-2314 was eluted with ethanol (1 mL) and mixed with sterile saline (9 mL). The purity and molar activity were determined by reverse-phase HPLC (InfinityLab Poroshell 120 PFP, 4.6 × 150 mm, 2.7 μm, Agilent) with UV and gamma detector connected in series. [$^{18}$F]OXD-2314 was eluted with acetonitrile-NH$_4$CO$_2$H (0.05 M) (50/50, v/v) at a flow rate of 3 mL/min (retention time = 4.5 min). Radiochemical identity was confirmed by co-injection of authentic standard with sample of final product formulation.

*(Pittsburgh)* No-carrier added [$^{18}$F]F$^-$ was produced via the $^{18}$O(p,n)$^{18}$F reaction (Siemens Eclipse HP cyclotron) and transferred in [$^{18}$O]H$_2$O to a lead-shielded hot cell where it was trapped on a Waters QMA light Sep-Pak (conditioned with 0.1 M NaHCO$_{3(aq)}$ (10 mL) followed by Milli-Q water (10 mL)), then eluted with K$_2$CO$_3$ (1.67 mg, 3.3 μmol) and Kryptofix 2.2.2 (4.2 mg, 1.5 μmol) in CH$_3$CN/H$_2$O (83:17, 1 mL) into a V-vial. The solvents were evaporated under argon flow at 110 °C, then the [$^{18}$F]fluoride was azeotropically dried with CH$_3$CN aliquots (1 mL × 2) at 110 °C under argon flow. Radiolabeling precursor 7 (2.5 mg) in DMSO (0.5 mL) was added, the V-vial was heated at 140 °C for 30 min, then cooled in an ice bath for 2 min. MeOH (0.25 mL) was added, the V-vial was heated at 140 °C for 20 min, then cooled in an ice bath for 2 min, and the reaction mixture was diluted with H$_2$O (20 mL). The solution was passed through a Waters tC$_{18}$ Sep-Pak (conditioned with EtOH (10 mL), then Milli-Q H$_2$O (10 mL)), then the crude radiotracer was eluted with CH$_3$CN (1 mL), collected in a vial, and purified by HPLC (Phenomenex Luna, 10 μm C18(2) 100 Å, 250 mm × 10 mm; 45:55 CH$_3$CN/0.1 M NH$_4$HCO$_3$ pH 4.6; 1 mL/min for 5 min, then 5 mL/min). The collected HPLC fraction was diluted with Milli-Q H$_2$O (50 mL), passed through a Waters tC$_{18}$ Sep-Pak (conditioned with EtOH (10 mL), then Milli-Q H$_2$O (10 mL)), and the Sep-Pak was rinsed with Milli-Q H$_2$O (10 mL). [$^{18}$F]OXD-2314 was eluted from the Sep-Pak with EtOH (1 mL), then 0.9% sterile saline (9 mL), passed through a 0.2 micrometer sterile filter, and collected in a sterile vial. The RCP (95%) and $A_m$ (0.32 GBq/nmol (8.76 mCi/nmol)) were determined by analytical HPLC (Phenomenex Luna C18(2), 5 μm, 250 mm × 4.6 mm; 45:55 CH$_3$CN/0.1 M NH$_4$HCO$_3$ pH 4.6; 2 mL/min). Radiochemical identity was confirmed by co-injection with authentic standard. The endotoxin level of the final formulation was <2.00 EU/mL (Charles Rivers Endosafe).

**Tritium-labeled radioligands.** Tritium-labeled compounds were custom-synthesized with high RCP and $A_m$: [$^3$H]OXD-2115: RCP > 96% and $A_m$ 1.21 ± 0.21 TBq/mmol (32.75 ± 6.02 Ci/mmol); [$^3$H]OXD-2314: RCP > 97% and $A_m$ 2.59 TBq/mmol (70.0 Ci/mmol); [$^3$H]PI-2620: RCP > 97% and $A_m$ 1.82 TBq/mmol (49 Ci/mmol); [$^3$H]Florzolotau: RCP 99% and $A_m$ 2.63 ± 0.35 TBq/mmol (70.5 ± 9.5 Ci/mmol); and [$^3$H]PiB: RCP > 98% and $A_m$ 3.01 TBq/mmol (81 Ci/mmol).

**Postmortem tissues–homogenates.** The binding assays utilized homogenates of fresh frozen, autopsy-confirmed, postmortem human AD, PSP, CBD, PiD, and control brain tissue blocks (1 cm³) obtained from the Neurodegenerative Disease Brain Bank at UCSF, which contained only frequent mixed 3 R/4R-tau neurofibrillary tangles and Aβ plaque aggregates (AD tissue, middle frontal gyrus), or only 4R-tau aggregates (PSP tissue, superior frontal gyrus or putamen/globus pallidus, or CBD tissue, middle frontal gyrus or putamen), or only 3R-tau aggregates (PiD tissue, middle frontal gyrus) and no other detectable aggregated amyloid species, or no detectable aggregated amyloid species at all (control tissue, middle frontal gyrus). Fresh frozen, autopsy-confirmed human PD anterior cingulate cortex brain tissue blocks (1 cm³) were obtained from Dr. Thomas Beach at Banner/Sun Health AZ with funding from the Michael J. Fox Foundation and contained frequent α-synuclein aggregates and no other detectable aggregated amyloid species. The frozen tissue blocks were prepared for the binding assays as previously described[47,48], except that equal quantities of the homogenates of three AD brains (AD1-AD3, Table S1) were pooled, as were homogenates from three PSP brains (PSP1-PSP3),

three CBD brains (CBD1-CBD3), three Pick's brains (PiD1-PiD3), and 4 PD brains (PD1-PD4) in order to help minimize the inter-subject variability within the different neuropathological groups. Individual subcortical brain tissues from a PSP subject (PSP4) and a CBD subject (CBD4), as well as with the middle frontal cortex of a control subject, were prepared for the binding assays as previously described[47,48].

**Equilibrium dissociation constant ($K_d$) assays.** Tritium-labeled radioligand binding assays utilized AD, PSP, CBD, PiD, PD, or PD brain homogenates to determine equilibrium dissociation constant ($K_d$) values and were performed as previously described[48,49].

**In vitro competition ($K_i$) assays.** The equilibrium inhibition constant ($K_i$) values of the unlabeled compounds were determined versus tritium-labeled radioligands using published methods[47].

**ARG (Toronto).** Fresh-frozen brains were cryo-sectioned (Leica CM3050S; 10 μm), thaw-mounted on glass slides (Fisher Superfrost Plus Gold) and stored at −80 °C for later use. [$^3$H]OXD-2314 (64 Ci/mmol) autoradiography was carried out where tissues were incubated with 3 nM radiotracer with 10 μM unlabeled OXD-2314 in 50 mM Tris/0.9% NaCl, 0.1% BSA, pH 7.4 for 90 min at room temperature. Sections were then washed with saline (3 × 5 min, 4 °C) and deionized water (diH$_2$O; 1 × 10 s, 4 °C) and air dried at room temperature. Slides were then exposed to phosphor screens (BAS- IP TR4020; GE Healthcare) for 6 days with a tritium standard for quantification (American Radiolabeled Chemicals, Inc.; St. Louis, USA). Images were generated with an Amersham Typhoon phosphorimager (GE Healthcare, USA). Region of interest (ROI) analysis was performed using MCID 7.0 imaging suite (Interfocus Imaging, Cambridge, UK) and data plotted using GraphPad Prism 8.

**IHC (Toronto).** Fresh-frozen human brain tissues were acclimated to room temperature and then exposed to post-fixation by acetone at −20 °C. Following a 10 min wash in buffer (tris-HCl buffered saline (TBS) containing 0.1% Triton-X), tissues were exposed to protein block (Vector 2.5% Normal Horse Serum) then incubated overnight at 4 °C with AT8 (MN1020; Invitrogen; 1:100 in protein block) antibody. Slides were washed with TBS followed by a 60 min incubation with ImmPRESS-AP horse anti-mouse IgG polymer. Following additional washes in TBS, sections were developed with a 5 min incubation in ImmPACT Vector Red alkaline phosphatase substrate reagent (Vector Laboratories). Slides were washed in TBS, followed by diH$_2$O, and cover-slipped in aqueous mounting media. Slides were imaged using an Olympus VS200 (Olympus Corporation, USA) at ×20 magnification.

**Autoradiography and immunohistochemistry (Sweden).** All tissues were kindly donated by Dennis Dickson from Mayo Clinic, USA and prepared as 4−5 μm slide mounted tissue sections. Sample size was determined by tissue availability. All slides underwent deparaffinization and Antigen retrieval (Ventana Discovery Ultra platform). This experimental was performed by Offspring Bioscience.

**ARG (Sweden).** Slides were washed and transferred to 1 phosphate-buffered saline. All sections were preincubated in binding buffer (50 mM Tris, pH 7.4) for 10 min at room temperature. [$^3$H]OXD-2314 autoradiography was carried out where tissues were incubated with 3 nM radiotracer with or without 1−3 μM unlabeled OXD-2314 in binding buffer for 40 min at room temperature at low pace shaking. Sections were then washed with buffer (3 × 10 min, 1 °C) and deionized water (diH$_2$O; 1 × 10 s, 4 °C) and air dried at room temperature. Slides were then exposed to phosphor screens (Fuji, BAS-TR2040) for 3 days with a tritium standard for quantification (Amersham Biosciences RPA506). Phosphoimage screens were processed using Image Reader FLA-7000 (IP Mode, LD650 Laser).

**IHC (Sweden).** IHC staining was made according to previous published method[50]. Reference antibody for Tau proteinopathies used were: Thermo Fisher Scientific (AT-8 Phospho-PHF-Tau). Briefly 5 μm slide mounted tissue sections was performed according to a standardized protocol with modifications to optimize the specificity of the observed staining patterns. IHC staining was performed using the Discovery Ultra platform (Ventana) automated immunostaining robot, using the OmniMap DAB chromogenic staining kit (Ventana Medical Systems) according to the manufacturer's instructions. In brief, initial deparaffinization, followed by heat activated antigen retrieval in a pH 8.0 buffer was performed to improve the detection of antigens in the FFPE tissue. Endogenous tissues peroxidases (Inhibitor CM, Ventana), which may interfere with the assays, were blocked with 3% hydrogen peroxide. The primary antibody for phosphorylated Tau detection AT-8 (MN1020, Thermo Fischer Scientific, 1:1000) was applied, followed by incubation with the HRP-conjugated secondary antibody (Omni-Map goat anti-mouse antibody, ready-to-use kit, Ventana). Visualization of the positively stained cells was performed by addition of hydrogen peroxide and DAB (single IHC) resulting in an insoluble brown (DAB) precipitate at the site of antibody binding. Counterstaining for IHC was done with hematoxylin (Hematoxylin II and Bluing Reagent, Ventana). The stained slides were subsequently scanned at up to 10 focal layers in brightfield (×20–40 objective) using a digital whole slide scanner (Panoramic 250 Scanner, 3D Histech, Budapest, Hungary). Analysis was performed of the IHC stained tissue sections by manual evaluation, using a digital image viewer software (CaseViewer).

**High-resolution ARG (Sweden).** All slides were run on the Ventana robot for deparaffinization, antigen retrieval and pTau AT-8 IHC as described under the IHC protocol section but by changing detection to turquoise (ChromoMap Teal, Ventana). After the completed Ventana run, slides were washed and transferred to 1XPBS before performing the radioligand incubation. All sections were preincubated in binding buffer (50 mM Tris, pH 7.4) for 10 min at room temperature. Tissues were incubated with 1–3 nM [³H]OXD-2314 with or without 1–3 μM unlabeled OXD-2314 in binding buffer for 40 min at room temperature at low pace shaking. Sections were then washed with buffer (3 × 10 min, 1 °C) and deionized water (diH2O; 1 × 10 s, 4 °C) and air dried at room temperature. In the dark, using a sodium lamp, ILFORD Nuclear Emulsion Type K2 emulsion (AGP9283, Oxford Instruments, Agar Scientific) was melted in heated water bath and diluted (1:1) with ddH2O. Emulsion was poured into dipping chambers, slides were dipped and placed vertically to air-dry. When completely dry, slides were placed in a light-tight slide box with desiccant and exposed at 4 °C for 1-12 weeks. After exposure, slides were developed in diluted Ilford Phenisol Developer (AGP9106, Oxford Instruments, Agar Scientific) for 2 min, rinsed in water and fixed in Ilford hypam fixer (AGP9183, Oxford Instruments, Agar Scientific) at 17 °C in a water bath. After rinsing in water, slides were counterstained with Harris HTX, dehydrated in graded ethanol, cleared in xylene, and mounted in Pertex. Images were taken using a 3D Histech P250 II scanner at up to 10 focal layers with a distance of 0.2 microns.

**PET/CT acquisition in rats, image analysis and dose assessment (Toronto).** A total of 8 healthy adult rats (Sprague-Dawley, 384 ± 56 g; 4/4 M/F) underwent PET imaging following bolus injection of [¹⁸F]OXD-2314 (20.5 ± 4.1 MBq), with two animals (M/F) scanned for 120 min with the head in the centre of PET field-of-view (FOV) whereas six animals (3/3 M/F) scanned for 180 min at three bed positions (head-to-bottom) with 15 frames (2×15 s, 2×30 s, 2×60 s, 1×120 s, 1×180 s, 2×240 s, 2×300 s, and 3×600 s) to cover the whole body. PET images were acquired using a nanoScan™ PET/CT scanner (Mediso, Budapest, Hungary) with rats anesthetized under isoflurane in O2 (1.5–2%, 1 L/min) throughout the imaging session and monitored closely for

body temperature and respiration rate. A scout CT was acquired for PET FOV positioning, then a material map CT image was acquired for PET corrections of attenuation and scatter and for PET/CT co-registration to define anatomic brain regions of interest. For the brain scans, aacquired list-mode data were sorted into thirty-nine three-dimensional (3D) (3×5 s, 3×15 s, 3×20 s, 7×60 s, 17×180 s, and 6×600 s), true sinograms (ring difference 84). The 3D sinograms were converted in 2D sinograms using Fourier rebinning with corrections for detector geometry, efficiencies, attenuation, and scatter before image reconstruction using 2D filtered back projection with a Hann filter at a cut-off of 0.50 cm⁻¹. The analytical reconstruction was also used for dose assessment of the multi-frame whole-body images. All data were corrected for dead time and were decay-corrected to the start of acquisition. Iteratively reconstructed images with the manufacturer's proprietary 3D algorithm (6 subsets, 4 iterations) were used for PET and CT co-registration, for co-registration of subject's CT with a stereotactic MR atlas of brain[51], for organ region-of-interest (ROI) placement, and for presentation in figures. Filtered back projection images were used to extract regional brain and organ time-activity curves (TACs) with VivoQuant® 2020 software (Invicro, Needham, MA, USA) and Amide (1.0.4)[52]. The selected organs (a total of 12) include brain, lungs, heart wall, heart content, liver, stomach content, spleen, kidneys, small intestine content, upper large intestine content, lower large intestine content, and urinary bladder content. Normalized cumulated activities of each organ were calculated using trapezoidal integration of the area under the TACs plus extrapolation of the TAC to infinity with natural decay of the radiotracer. Dose assessment was performed with OLINDA/EXM 1.1 to calculate the effective dose and the effective dose equivalent. SUV were calculated by normalizing regional radioactivity for injected radioactivity and body weight.

**Rat radiometabolite analysis (Toronto).** Five adult Sprague-Dawley rats (3 F/2 M, 378 ± 51 g) were used in [¹⁸F]OXD-2314 radiometabolite analysis. Following injection of 46.7 ± 3.3 MBq of the radiotracer through a tail-vein catheter, rats were sacrificed by decapitation 30 min (n = 2) and 60 min (n = 3) post-injection. Trunk blood was collected and centrifuged for 5 min at 2500 × g, and the plasma was collected. Brains were quickly extracted and homogenized with a BeadBug™ (Benchmark Scientific; Sayreville, NJ, USA) in 3.2 mL acetonitrile together with 1 μg of unlabeled parent compound, then 1.6 mL deionized H2O was added and mixed. The homogenate was centrifuged at 12,000 × g for 5 min, and the supernatant was collected. Parent radiotracer was separated from metabolites in plasma and brain extract by column-switching HPLC in a mobile phase of 40/60 CH3CN/ 100 mM NH4HCO2 (v/v) and analyzed with PowerChrom 2.6.15 (eDAQ).

**PET/CT acquisition in NHP (Pittsburgh).** NHP imaging studies in a male (13 kg) macaca mulata monkey and male (25.7 kg) and female (14.2 kg) papio anubis baboons were carried out in accordance with the guidelines set forth by the Institutional Animal Care and Use Committee at the University of Pittsburgh. Additional details regarding the maintenance of an active environmental enrichment program for the housed NHPs are described in Supplemental Information. Dynamic [¹⁸F]OXD-2314 PET imaging of the brains of the isoflurane anesthetized NHPs was performed using a Siemens Biograph mCT Flow 64-4 R PET/ CT scanner (22.1 mm axial FOV, maximum intrinsic spatial resolution of 4.3 mm FWHM[53]. Following a low-dose CT scan for attenuation correction of PET emission data, [¹⁸F]OXD-2314 was injected intravenously and emission data were acquired in list-mode for a period of 90 min. Emission data were binned into a dynamic series of 32 frames (20 sec-10 min) and reconstructed using filtered back projection with Fourier rebinning (FORE + FBP) with standard corrections applied that included attenuation, scatter, electronics dead-time, and physical decay.

**NHP radiometabolite analysis (Pittsburgh).** Venous blood samples (3 mL) were collected at 2, 10-, 30-, 45-, and 90-minutes following radiotracer administration. The blood samples were transferred from the sample syringe into centrifuge tubes and centrifuged (Eppendorf MiniSpin 5452) for 45 s at $11,300 \times g$. The separated plasma (~0.5 mL) was transferred to a centrifuge tube and diluted with an equal volume of acetonitrile. The resultant solution was vortexed for 1 min and then centrifuged for 45 s at 12,500 rpm. The resulting supernatant was analyzed by reverse-phase HPLC (Phenomenex Luna C18(2), 5 µm, 250 mm × 4.6 mm; 45:55 $CH_3CN$/0.1 M $NH_4HCO_3$ pH 4.6; 2 mL/min) and analyzed using a Raytest 3" × 3" sodium iodide detector with pinhole flow cell (2.0 mL) and Gabi GinaStar software (version 4.07). System suitability of the analytical system was evaluated by injection of an authentic standard of OXD-2314 using the same analytical system described above using a Waters UV detector (Model 481, 346 nm).

**Statistics and reproducibility.** Sample sizes for in vitro studies were determined by the availability of human tissue samples. Sample sizes for using animals were guided by ethical reasoning to minimize the number of animals used. No data were excluded from the analyses.

### Reporting summary

Further information on research design is available in the Nature Portfolio Reporting Summary linked to this article.

## Data availability

Data source files for all figures and tables are provided with the submission. Any additional data is available upon request. Source data are provided with this paper. Additional figures, tables, synthetic and radiochemical methods, and compound characterizations are available in supplementary information.

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

## Acknowledgements
S.S., C.A.M., and N.V. thank the Rainwater Charitable Foundation (Tau Consortium), CurePSP and the Michael J. Fox Foundation for Parkinson's Research (MJFF-024347 & MJFF-024348) for jointly supporting this research collaboration. N.V. also thanks the Azrieli Foundation, Canada Foundation for Innovation, Ontario Research Fund and the Canada Research Chairs Program for support. Human brain tissue samples were provided by the Neurodegenerative Disease Brain Bank at the University of California, San Francisco (which receives funding support from NIH grants P30AG062422, P01AG019724, U01AG057195, and U19AG063911, the Rainwater Charitable Foundation, and the Bluefield Project to Cure FTD) and by the Banner/Sun Health Brain Bank funded by the Michael J. Fox Foundation. Tissue for autoradiography was obtained from the Douglas Bell Canada Brain Bank or Dennis Dickson at the Mayo Clinic.

## Author contributions
A.L., E.M., J.T., N.S.M., D.S., J.S., P.S., J.S.S., B.J.L., and J.V. performed research; A.L., E.M., J.T., D.S., N.S.M., J.S.S., B.J.L., S.S., C.A.M., and N.V. analyzed the data; A.L., E.M., C.A.M., and N.V. wrote the paper.

## Competing interests
S.S., D.S., and J.V. were employees of Oxiant Discovery during the conduct of the reported study who may own or hold stock options in the company. N.V. is a co-founder of MedChem Imaging, Inc., which did not contribute support for this study. The remaining authors declare no competing interests.
