## [Peer Review File · Nature Communications]

Ligand-based design of [18F]OXD-2314 for PET imaging in non-Alzheimer's disease tauopathiesREVIEWER COMMENTS

Reviewer #1 (Remarks to the Author):

This manuscript nicely describes the discovery and preclinical characterization of a possible novel 18F radiotracer for the detection of 4R Tau pathology in the human living brain. This remains an unmet need in the field, despite some of the next generation Tau PET tracers, such as Florzolutau and PI-2620, have already shown initial promising clinical data.

Overall, the medicinal chemistry strategy for the improvement of the previously identified candidate, OXD-2115, is well exemplified and the studies describing the preclinical characterization of the newly identified candidate, OXD-2314, are well conducted.

However, OXD-2314 is shown not be selective for 4R Tau inclusions but rather to be, at least in vitro, a good Tau binder also to the 3R/4R Tau aggregates (e.g. AD) and to the 3R Tau aggregates (e.g. PiD). The abstract and the discussion should therefore be modified to emphasize more this aspect.

Moreover, before considering this manuscript for publication, additional preclinical data on OXD-2314 should be provided, particularly:

1) Autoradiography data in human fresh frozen tissues from different Tauopathies (AD, PSP, CBD, and PiD) as well as from non-diseased controls and PD. This should be complemented by immunohistochemistry data with Tau, Abeta and a-syn antibodies and ideally by nuclear emulsion autoradiography data to show target engagement of 3H-OXD-2314 to the specific Tau inclusions present in 4R Tau pathology (e.g. tufted astrocytes).

2) Table 3 should be complemented with Bmax data for each tested condition

3) Data on the selectivity of OXD-2314 versus amyloid beta should be presented (and not only cited in the text)

4) Selectivity of OXD-2314 versus MAO-A and MAO-B should be proved not only with enzymatic assays, as in S8, but also in binding assays e.g. with specific inhibitors such as clorgyline and deprenyl

Additionally, the manuscript will benefit if the following points are addressed:

1) the legend in table 2 should clarify if replicates are from single or multiple donors. In case they are from a single donor, data from additional donors should also be provided.

2) In the NHP PET scan, SUV curves for additional brain regions relevant e.g. for PSP should be reported. Additionally, an in vivo blocking study in NHP, should be considered to assess whether the signal remaining at 90 min post-injection is due to a specific off-target binding or unspecific retention. Lastly,

the PET scan could be repeated in a female monkey to assess whether the slower metabolism observed in female vs male rats is also seen in NHP, with a possible impact on the signal to noise ratio in the human brain.

3) in S1, deriving AUC for OXD-2115 with data from only 2 time points is very imprecise and should be avoided, particularly because the C_{max} is not clearly defined. Instead the brain to plasma ratio should be indicated at one or two selected time points for both compounds. Unbound fractions of both compounds in plasma and brain should also be indicated.

4) in the method section for the radiobinding assays, the use of total brain homogenates or of a specific biochemical fractions should be clearly indicated

5) additional cross-competition studies could be added to provide more information on the possible binding site of OXD-2314 (e.g. competition of OXD-2314 with MK-6240 in AD and with Florzolotau in PiD)

Reviewer #2 (Remarks to the Author):

This study assessed the utility of a new potential tau PET ligand, [18F]OXD-2314, as a high affinity ligand for binding to 4-repeat (4R) tau. The authors identified OXD-2314 by assessing the in vitro tau affinity of over 150 analogues of OXD-2115. They showed that OXD-2314 had higher affinity to tau aggregates in Alzheimer's disease, progressive supranuclear palsy (PSP) and corticobasal degeneration (CBD) than three other ligands (PI-2620, florzolotau and OXD-2115) and demonstrated that OXD-2314 binds to a different high affinity site on tau aggregates compared to PI-2620 and florzolotau. They also found no off-target binding to MAO-A or MAO-B. Suitably high brain penetration and rapid clearance properties were identified in mice studies. The authors conclude that OXD-2314 is ready to be translated for first-in-human PET studies.

This work is very high impact with many laboratories and researchers wanting a tau PET ligand to investigate 4R tauopathies. The studies presented in this manuscript are essential steps to allow the use of OXD-2314 in humans. I do not have any major concerns, but have some comments/questions for the authors to address:

1. The tissue sample for the PSP patients came from the superior frontal gyrus, rather than brainstem or subcortical regions which would be expected to have the greatest tau burden in a typical PSP case. Was the pathology in the PSP and CBD case/cases atypical? Was the clinical diagnosis of the PSP case/s PSP and for CBD was it CBS? That is, were these typical PSP and CBD cases?

2. Very little information is provided on the 4R tau cases used in this study. How many PSP, CBD, PiD and AD cases were included? What was the Braak NFT stage of the PSP, CBD and PiD cases. This is important information to determine whether the 4R tau cases had co-existing 3R+4R tau pathology that the ligand could be binding.

3. Radioactivity washout was slower in the female rats (with higher overall brain radioactivity) compared

to the male rats – could the authors discuss this finding and whether this may have implications for sex differences in humans.

4. It is very useful that the authors assessed binding to MAO-A and MAO-B, but it would be very important to also assess binding to other non-tau targets that are a problem in 4R tauopathies, such as iron in the basal ganglia and neuromelanin in the substantia nigra.

5. It would be useful for the authors to show a picture of the uptake in a PSP and or CBD case compared to the uptake in an AD case, even if in the superior frontal lobe. This is to get a sense of whether the signal is PSP (or CBD) is truly any different from what we are seeing with flortaucipir and the other ligands and how different it is from what occurs in AD.

Reviewer #3 (Remarks to the Author):

The manuscript by Lindberg et al. describes their efforts to develop a 4R-tau PET tracer. The authors have previously reported a lead, [18F]OXD-2115 that showed good affinity toward 4R, but limited brain uptake. Their submitted manuscript describes the work toward optimizing this previous lead compound with regard to brain permeability. Around 150 compounds were explored and screened for tau affinity, and from these [18F]OXD-2314 was selected as the most promising candidate for further evaluation. The topic is timely, highly relevant and nicely introduced, describing the importance of a tracer for non-AD tauopathies. However, there are some aspects of the work for which further information could be helpful for the readers. One major comment; the aim was to develop a ligand selective toward 4R-tau, but the selected candidate binds with high affinity to both 3R and 4R-tau. I believe this has to be to be stronger acknowledged in the manuscript. Also, the title of the manuscript indicates a selective 4R-tau PET tracer. Nevertheless, the tracer has high affinity for non-AD tauopathies and the presented work can stimulate further advances in the field. Below are some of the concerns and question I have on the current version of the manuscript.

Major comments

1) The title of the manuscript “Structure-guided design of pyridinyl-indole 4R-tau PET radiotracers: Development of [18F]OXD-2314 for human use” implies;

- that the authors made use of a protein structure/binding mode of the lead during the design phase. The term is also used in the manuscript (page 5 line 77). Consider using the term ligand-based design instead.

-that the tracer is selective toward 4R-tau over 3R-tau. However, the presented data shows that the tracer has a high affinity for 3R-tau as well. The authors also state this on page 12 lines 201-209.

Consider changing "4R-tau" to "3R/4R-tau" or "tracer for non-AD tauopathies". The same is with the abstract line 21.

2) Page 5 line 79: “Lead candidates were further evaluated against [3H]OXD-2115 in human postmortem AD, CBD and PSP tissues for comparison with first- and second-generation PET radiopharmaceuticals for imaging tau aggregation.”

What candidates were tested on postmortem tissues? Only data for the selected tracer candidate is

shown in table 2 and 3.

3) The authors synthesized more than 150 compounds with R-group exploration in two different positions, with the aim to improve the physicochemical properties for brain uptake. It is not clear from the text if they made use of the in silico prediction scores (BBB and CNS (PET) MPO scores) already at this stage, or if scoring was only performed for the few selected leads presented in table 1?

Why not include the structures/affinity and in silico predictions scores for these ~150 compounds as a table in the supporting information? The SAR can also be summarized in a figure instead of the current scheme 1, along with (or alternatively) graphs displaying any trends in properties (e.g., BBB and MPO scores, HBDs, lipophilicity, TPSA, etc) vs. affinity. This is extensive work and an important part of the project, but it's given too little attention in the current state of the manuscript.

On a small note, the first screening of the ~150 compounds, was this displacement at a single point concentration or directly full binding curves for K_i determination?

4) Page 6 line 100-101: "Two enantiomeric analogues of OXD-2115, namely, OXD-2188 and OXD-2189, were initially identified as promising candidates."

Are these the two enantiomers of the initial lead compound OXD-2115 or enantiomers of a structural analog of OXD-2115? I assume it is the latter since on page 9 line 147-148, the authors write that these two tracers had lower lipophilicity than OXD-2115, which would not be the case if they are enantiomers. Please clarify this in the first sentence, and add the structures to the main manuscript or in the supporting information (combined with the presented PET data). In addition, assuming these compounds were enantiomers of a structural analog of OXD-2115: What was the affinity in vitro and was it different between the enantiomers? Even though they have similar in vitro affinity the PK, ADME(T), etc can of course differ, which justifies evaluating both in vivo. However, since the structures are not specified it does not add much to the manuscript. Either add more information regarding these compounds, including structures or consider to take this out.

5) It is not totally clear why OXD-2314 was selected as the most promising lead. The K_i values presented in table 1 are from $n=1$, which is fine for a preliminary screening. However, the last 5 compounds in table 1 have very similar profile. Did the in silico prediction scores of these compounds differ a lot? Were those scores used to prioritize the compounds? Adding columns with the scores for each compound to the table would be helpful. In addition, a sentence about your criteria and thresholds for affinity in the different tissue homogenates would be good for clarification. Do you want equally potent and/or higher in AD tissue?

6) The section called "BBB-permeability": See earlier comment if this section can be combined with the lead identification. An extensive description and discussion of the in silico prediction scores are given. These are tools to assist in the design and/or to prioritize compounds. As it is now, too much attention is given to these. This text can be shorten. More importantly, the threshold stated for high probability of brain permeability in each model is incorrect (Page 9 line 140-142). The threshold in each model for high likelihood of success is as follows: >4 (CNS MPO), >3 (CNS PET MPO), and >4 (BBB score). Meaning your candidate is slightly below the threshold in all scores.

Please keep in mind when discussing your result (page 9-10, line 153-158 + discussion) that these scores do not predict the same. The BBB score only predict passive permeability, so a score above 4 means that a compound most likely cross the BBB. Whereas, the CNS MPO scores not only take into account permeability, but also ADME(T) properties (for tracers e.g., non-specific binding, p-gp liability), meaning you predict the likelihood of overall success as a CNS drug and PET tracer, respectively.

7) Information about the software used for calculating the properties has to be provided. The final score will differ depending on the software used for calculating the properties, especially for ClogP, pKa and ClogD7.4. For example, the CNS (PET) MPO is trained on descriptors from ChemDraw, Biolum and ACD Percepta. ChemAxon and MOE work fine with the BBB score.

If the authors have used ChemAxon, which has rather good correlation with the original software for CNS MPO prediction, I refer the authors to a paper from 2021 by Urbina et al. (ACS Chem. Neurosci. 2021, 12, 12, 2247–2253).

8) Not enough details are provided regarding the in vitro binding experiments. The authors refer to references 43 and 44 in the manuscript. I assume the assays have been performed similarly, but then this should be stated and the concentration ranges for each compound, amount of the different homogenates, conditions, etc for the assays in the present work should be added. Maybe the authors can also refer to their previous paper in ACS Chem Neurosci from 2021?

In addition, how is the non-specific binding assessed in the experiments determining Kd of the 3H-ligands on different tissues? Self-blocking?

9) No experimental details are provided for the PK studies in mice.

10) Page 13, line 221-224 “Of note, OXD-2314 was found to inhibit the binding of the antagonist radioligand (protriptyline) to the norepinephrine transporter(h) by 62.9% and the enzyme activity of 5-lipoxygenase by 50.”

How is the target abundance and regional expression levels of these off-targets compared to tau in the disease specific conditions the authors want to assess? If known, this information is worth adding, as well as if the authors foresee any potential issues here?

11) Why haven't the authors performed any studies in an animal model with tau filaments? For example, the 4R-tau expressing P301L mice?

Minor issues

Page 11 line 179: left square bracket missing, [3H]PI-2620

Page 14 line 251: delete “PET” in figure legend “...of female and male rat PET brain...”

REVIEWER COMMENTS

Reviewer #1 (Remarks to the Author):

This manuscript nicely describes the discovery and preclinical characterization of a possible novel ¹⁸F radiotracer for the detection of 4R Tau pathology in the human living brain. This remains an unmet need in the field, despite some of the next generation Tau PET tracers, such as Florzotau and PI-2620, have already shown initial promising clinical data.

Overall, the medicinal chemistry strategy for the improvement of the previously identified candidate, OXD-2115, is well exemplified and the studies describing the preclinical characterization of the newly identified candidate, OXD-2314, are well conducted.

However, OXD-2314 is shown not be selective for 4R Tau inclusions but rather to be, at least in vitro, a good Tau binder also to the 3R/4R Tau aggregates (e.g. AD) and to the 3R Tau

aggregates (e.g. PiD). The abstract and the discussion should therefore be modified to emphasize more this aspect.

We thank the Reviewer for their support of our manuscript. We agree with the Reviewer regarding the distinction of [¹⁸F]OXD-2314 as a 4R-tau PET radiotracer and we have updated the manuscript to reflect that point and revised the title so that the emphasis is now on non-Alzheimer's disease tauopathies as follows:

Title is revised as follows:

"Ligand-based design of pyridinyl-indole PET radiotracers for non-Alzheimer's disease tauopathies: Development of [¹⁸F]OXD-2314 for human use"

Abstract is revised as follows:

"To date, no high-affinity non-AD tau- PET radiopharmaceutical has been optimized for imaging non-AD tauopathies."

Introduction page 5 now includes the following text:

"In the present study, novel pyridinyl-indole analogues of OXD-2115 were identified using ligand-based design to select for high potency and selectivity for non-AD tau aggregates as well as improved brain uptake.

Results, *In vitro* binding assay section is revised as follows:

"Overall, [³H]OXD-2314 was shown to be a high affinity ligand for tau aggregates in both AD and non-AD tauopathies.";

and:

"It is noteworthy that [³H]OXD-2115 showed high affinity to the primarily 4R-tau (PSP, CBD) aggregates in vitro but had lower affinity to the predominantly 3R-tau aggregates in PiD, and [³H]OXD-2314 was expected to be similar, but instead showed high affinity for all of the non-AD and mixed 3R/4R-tauopathies (AD) in human brain homogenate affinity assays (Table 2)."

Discussion is revised as follows:

"These results indicate that [³H]OXD-2314 is a selective high-affinity tau PET radiotracer with unique binding sites on tau aggregates for non-AD tauopathies."

Moreover, before considering this manuscript for publication, additional preclinical data on OXD-2314 should be provided, particularly:

1) Autoradiography data in human fresh frozen tissues from different Tauopathies (AD, PSP, CBD, and PiD) as well as from non-diseased controls and PD. This should be complemented by immunohistochemistry data with Tau, Abeta and a-syn antibodies and ideally by nuclear emulsion autoradiography data to show target engagement of 3H-OXD-2314 to the specific Tau inclusions present in 4R Tau pathology (e.g. tufted astrocytes).

The Authors acknowledge that no autoradiography was performed as it was not seen as essential for making a decision on translating [¹⁸F]OXD-2314 for human application, given the results from

the reported *in vitro* data in the manuscript. However, in light of the Reviewer's comment, a new batch of [³H]OXD-2314 was synthesized and autoradiography was performed at two separate laboratories in AD and PSP or CBD tissue and compared to healthy controls and tau immunohistochemistry. In addition, high resolution autoradiography was performed using nuclear emulsion to exhibit co-localization of autoradiographic signal and phospho-tau immunohistochemistry. Additional text in the manuscript has been added and refers to the revised Supporting Information (Figures S21-S23 and Table S5) as follows:

"Autoradiography studies were also conducted with [³H]OXD-2314 in healthy control, AD, CBD and PSP tissues for total radiotracer signal and blocking with unlabeled OXD-2314, and compared to AT8 immunohistochemistry for tau staining. Signal was displaceable under homologous blocking conditions to the level of nonspecific binding present in healthy control tissue and elevated radiotracer signal was observed in AD and PSP hippocampal tissues compared to control. High resolution autoradiography with [³H]OXD-2314 using nuclear emulsion aligned with staining for phospho-tau deposits demonstrated by IHC in cortical PSP tissue (Figures S21-S23 and Table S5)."

Autoradiography using [³H]OXD-2314

Figure S21. [³H]OXD-2314 binding in HC, AD and PSP hippocampal tissue. Total [³H]OXD-2314 (3 nM) binding is shown along with displacement by unlabeled OXD-2314 (10 μM) compared to AT8 immunostaining for tau. Increased radiotracer signal was observed in AD and PSP tissue compared to healthy control. Radiotracer signal in AD and PSP tissue was displaced by homologous blockade and was highest in the AD cases.

Figure S22. [³H]OXD-2314 binding in CBD and PSP cortical tissue. Total [³H]OXD-2314 (3 nM) binding is shown along with displacement by unlabeled OXD-2314 (3 μM) compared to AT8 immunostaining for tau.

Figure S23. $[^3\text{H}]$ OXD-2314 high resolution autoradiography in CBD cortical tissue. $[^3\text{H}]$ OXD-2314 signal (1 or 3 nM) (black) is overlaid with AT8 staining for phospho-tau (turquoise).

Table S5. Tissue Demographics used in autoradiography with [³H]OXD-2314.

Primary Pathological Diagnosis	Case ID	Region	Sex	Age
HC; A0B1C0; Low NFT	HC1	BA28/HP	F	72
Healthy	HC2	BA28/HP	F	51
AD; A3B2C3	AD1	BA28/HP	M	90
AD; A3B3C3	AD2	BA28/HP	M	88
AD; A3B3C3	AD3	BA28/HP	F	93
AD; Braak IV/V	AD4	BA28/HP	F	96
PSP	PSP1	BA28/HP	M	65
PSP	PSP2	BA28/HP	F	72

*BA: Brodmann's area; M: male; F: female

2) Table 3 should be complemented with Bmax data for each tested condition

We thank the Reviewer for the suggestion to include Bmax for the human tissues to complement Table 3 (now renumbered as Table 4). Additional experiments have been completed and Bmax values for [³H]OXD-2314 in AD, PSP, CBD, PiD and PD are now included as a new Table (Table 3 in revised manuscript), and additional text is provided below:

“The B_{max} for [³H]OXD-2314 in AD, PSP, CBD and PD frontal cortex tissue was measured showing highest B_{max} value in AD tissue with 430 ± 110 nM, comparable values in both PSP and CBD in both frontal cortex and subcortical cortex ranging from 96 to 170 nM, the lowest value was found in PiD tissue with 63 ± 21 nM and in PD, [³H]OXD-2314 measured a B_{max} value of 240 nM (Table 3).”

Table 3. B_{max} values for [³H]OXD-2314 in AD, PSP, PiD and PD tissue.

Radiotracer	Brain region	AD (n = 3) B _{max} (nM)	PSP (n = 3) B _{max} (nM)	CBD (n = 3) B _{max} (nM)	PiD (n = 3) B _{max} (nM)	PD (n = 1) B _{max} (nM)
[³ H]OXD-2314	Frontal cortex	430 ± 110	130 ± 50	96 ± 32	63 ± 21	240
	Subcortical	-	170 ± 40	110 ± 30		

3) Data on the selectivity of OXD-2314 versus amyloid beta should be presented (and not only cited in the text)

We thank the Reviewer for this suggestion. We have now added the data for the OXD-2314 competition assay against amyloid tracer [³H]PiB which is included in revised Table 4 as shown below:

Table 4. Inhibition constant (K_i) values (n = 1) of unlabeled OXD-2115, PI-2620, florzolotau, tauvid, MK-6240 and OXD-2314 against [³H]OXD-2314 and unlabeled OXD-2314 against [³H]OXD-2115 in AD, PSP, and CBD tissue. * n = 3

	Competitor	AD K _i (nM)	PSP K _i (nM)	CBD K _i (nM)	PiD K _i (nM)
[³ H]OXD-2314	OXD-2115	14	27	30	23
	PI-2620	460	620	660	760
	Florzolotau	>10,000	4200	>10,000	>10,000
	Tauvid	>10,000	>10,000	>10,000	4400
	MK-6240	1900	5600	1700	1000

	OXD-2314	4.8 ± 0.3*	5.8 ± 0.2*	6.5 ± 0.4*	5.5 ± 0.5*
[³H]OXD-2115	OXD-2314	8.8	4.8	4.4	-
[³H]PiB	OXD-2314	161 ± 34*	-	-	-

4) Selectivity of OXD-2314 versus MAO-A and MAO-B should be proved not only with enzymatic assays, as in S8, but also in binding assays e.g. with specific inhibitors such as clorgyline and deprenyl.

Additional in vitro binding studies were performed based on the Reviewers suggestions using deprenyl (MAO-B) and Ro41-1099 (MAO-A) to measure K_i values in AD, PSP and CBD tissue. Table S4 has been added to Supporting Information showing K_i values of [³H]OXD-2314 against MAO-A and MAO-B antagonists deprenyl and Ro41-1049.

Table S4. Blocking study using [³H]OXD-2314 in AD, PSP and CBD tissue against MAO-A and MAO-B substrates deprenyl and Ro41-1049.

Radioligand	Cold Compound	Tissue	K_i (nM)
[³ H]OXD-2314	Deprenyl	AD ($n = 1$)	2900
[³ H]OXD-2314	Deprenyl	PSP ($n = 1$)	3000
[³ H]OXD-2314	Deprenyl	CBD ($n = 1$)	5000
[³ H]OXD-2314	Ro41-1049	AD ($n = 1$)	>10,000
[³ H]OXD-2314	Ro41-1049	PSP ($n = 1$)	>10,000
[³ H]OXD-2314	Ro41-1049	CBD ($n = 1$)	>10,000

Additionally, the manuscript will benefit if the following points are addressed:

1) the legend in table 2 should clarify if replicates are from single or multiple donors. In case they are from a single donor, data from additional donors should also be provided.

Complete data for the human tissues that were used has been added to SI as **Table S1**, as follows:

Table S1. Summary of demographic and neuropathologic characteristics of donors and postmortem brain tissues. Key to scores: 1 = mild, 2 = moderate, 3 = severe; or 1 = small, 2 = moderate, or 3 = large numbers of given hallmark.

Case	Sex	Age at death	Clinical Diagnosis	Neuropathology Diagnosis	PMI (hrs)	Frozen Block Anatomical Location	Abeta_CERAD_Neuritic plaques	Abeta_Diffuse Plaques	Abeta_CAA	CP13_Neurofibrillary tangles
AD1	M	63	AD	AD	6.0	Middle/inferior frontal gyrus, anterior	3	3	0	3
AD2	M	64	AD	AD	6.8	Middle/inferior frontal gyrus, anterior	3	3	1	3
AD3	F	66	AD	AD	15.4	Middle/inferior frontal gyrus, anterior	3	3	2	3
PSP1	M	63	PSP	PSP	4.4	Superior frontal gyrus	0	0	0	0
PSP2	F	70	PSP	PSP	7.2	Superior frontal gyrus	0	0	0	0
PSP3	M	68	PSP	PSP	10.1	Superior frontal gyrus	0	0	0	0
PSP4	F	66	PSP	PSP	9.2	Putamen & Globus Pallidus	0	0	0	0
CBD1	M	68	PNFA	CBD	7.4	Middle/inferior frontal gyrus, anterior	0	0	0	0
CBD2	F	70	CBS	CBD	7.4	Middle/inferior frontal gyrus, anterior	0	0	0	0
CBD3	F	63	CBS	CBD	5.0	Middle/inferior frontal gyrus, anterior	0	0	0	0
CBD4	M	75	CBS	CBD	7.4	Putamen	0	0	0	0
PiD1	M	64	CBS	Pick's	13.7	Middle/inferior frontal gyrus, anterior	0	1	0	0
PiD2	F	63	PNFA	Pick's	4.3	Middle frontal gyrus, anterior	0	0	0	0
PiD3	F	69	bvFTD	Pick's	6.3	Middle/inferior frontal gyrus, anterior	0	0	0	0
PD1	M	67	PD	PD	4.2	Anterior cingulate	0	0	0	0
PD2	F	72	PD	PD	6.1	Anterior cingulate	0	0	0	0
PD3	F	73	PD	PD	7.3	Anterior cingulate	0	0	0	0
PD4	M	69	PD	PD	5.3	Anterior cingulate	0	0	0	0
Control	F	86	Control	Control	6.4	Middle/inferior frontal gyrus, anterior	0	0	0	0

Case	CP13_Pick bodies	CP13_Neuronal cytoplasmic inclusions	CP13_Globose tangles	CP13_Astrocytic plaques	CP13_Tufted astrocytes	CP13_Thorny astrocytes	CP13_Neuropil threads/grains	CP13_White matter threads/grains	CP13_Glial cytoplasmic inclusions	aSyn_Lewy Bodies	aSyn_Lewy Neurites	aSyn_Glial cytoplasmic inclusions
AD1	0	0	0	0	0	0	3	1	0	0	0	0
AD2	0	0	0	0	0	0	3	1	0	0	0	0
AD3	0	0	0	0	0	0	3	1	0	0	0	0
PSP1	0	3	0	0	3	1	2	2	3	0	0	0
PSP2	0	2	0	0	3	1	1	1	3	0	0	0
PSP3	0	2	1	0	3	1	1	0	2	0	0	0
PSP4	0	2	1	0	3	0	1	1	3	0	0	0
CBD1	0	3	0	3	0	0	3	3	2	0	0	0
CBD2	0	3	0	3	0	1	3	3	3	0	0	0
CBD3	0	2	0	3	0	0	2	2	1	0	0	0
CBD4	0	3	0	3	0	0	3	3	3	0	0	0
PiD1	3	1	0	0	0	0	3	1	3	0	0	0
PiD2	3	1	0	0	0	0	2	2	3	0	0	0
PiD3	3	1	0	0	0	0	3	2	3	0	0	0
PD1	0	0	0	0	0	0	0	0	0	3	3	0
PD2	0	0	0	0	0	0	0	0	0	2	3	0
PD3	0	0	0	0	0	0	0	0	0	3	3	0
PD4	0	0	0	0	0	0	0	0	0	3	3	0
Control	0	0	0	0	0	0	0	0	0	0	0	0

We have also edited the Legend of Table 2 which now reads as follows:

Table 2. *In vitro* equilibrium dissociation constant (K_d) values ($n = 3$; mean \pm SD) of [3 H]PI-2620, [3 H]florzolotau, [3 H]OXD-2115, and [3 H]OXD-2314 in AD, PSP, CBD, PiD, and PD frontal cortex tissues as well as P301L transgenic mouse tissue (see Table S1 for demographics and neuropathology of subjects). (n.d. = not determined; undefined = insufficient specific binding to fit the data). The AD, PSP, CBD, and PiD homogenates were pooled from 3 cases, the PD homogenates were pooled from 4 cases, and the Control was a representative single subject. * Subcortical tissue

2) In the NHP PET scan, SUV curves for additional brain regions relevant e.g. for PSP should be reported. Additionally, an in vivo blocking study in NHP, should be considered to assess whether the signal remaining at 90 min post-injection is due to a specific off-target binding or unspecific retention. Lastly, the PET scan could be repeated in a female monkey to assess whether the slower metabolism observed in female vs male rats is also seen in NHP, with a possible impact on the signal to noise ratio in the human brain.

We completed additional PET studies in NHP in order to answer the Reviewers questions. One additional macaque PET scan with displacement of unlabeled OXD-2314 was performed to evaluate any potential specific off target binding and the potential sex differences were evaluated in male and female baboons as discussed and seen below:

“Dynamic PET imaging was performed in a rhesus macaque (*macaca mulatta*) monkey following the injection of [¹⁸F]OXD-2314 at baseline (104.9 MBq, *n* = 1) and was repeated in a second [¹⁸F]OXD-2314 study (115.8 MBq, *n* = 1) where a displacing dose of unlabeled OXD-2314 (0.5 mg/kg) was administered 10 min after [¹⁸F]OXD-2314 injection to verify the absence of specific binding in NHP brain. Venous blood samples were drawn to characterize the metabolic fate of [¹⁸F]OXD-2314 in the macaque. Dynamic [¹⁸F]OXD-2314 PET imaging was also performed in a male (106.6 MBq, *n* = 1) and a female (115.8 MBq, *n* = 1) baboon (*papio anubis*), and venous blood samples were drawn in both baboons to characterize potential effects of sex on radiometabolism. Bilateral regions of interest were defined on 5-8 contiguous axial image planes for the lateral temporal cerebral cortex, cerebellum, striatum, and whole brain, which were used to sample dynamic [¹⁸F]OXD-2314 PET images to determine the time-varying radioactivity concentration expressed in SUV units. [¹⁸F]OXD-2314 readily entered macaque brain (whole brain peak ~2.2 SUV at ~2 min) and distributed uniformly throughout cortical and subcortical grey matter regions (**Figure 3A**). Radioactivity cleared rapidly from the whole brain with a clearance half-time of ~22 min and resulted in a 2:90 min radioactivity concentration ratio of 3.9 (**Figure 3B**). No preferential retention of [¹⁸F]OXD-2314 in any brain region was observed, suggesting that signal in NHP brain was dominated by non-displaceable radioactivity (non-specifically bound and free radiotracer). Administration of a pharmacologic dose of unlabeled OXD-2314 (0.5 mg/kg) 10 min after [¹⁸F]OXD-2314 injection (**Figure 4**) resulted in a very small transient increase in brain radioactivity concentrations, possibly resulting from an increase in plasma concentrations of free [¹⁸F]OXD-2314 released from peripheral sites. There was no evidence of accelerated brain clearance in any brain region that would suggest a significant component of displaceable (specifically bound) [¹⁸F]OXD-2314 to any site in normal monkey brain. These data are consistent in showing that non-displaceable binding is the only significant source of [¹⁸F]OXD-2314 signal in normal NHP brain. Radiometabolite analysis of macaque venous blood identified only one, polar radiometabolite, which increased from 37% at 10 min to 77% at 90 min post injection and was not likely brain penetrant (**Figures S19-S20**).

Figure 3. A) PET summation images (0-11 min and 60-90 min) and B) time-activity curves of whole brain (●), cortex (▲), and cerebellum (○) from NHP PET imaging of $[^{18}\text{F}]\text{OXD}-2314$.

Figure 4. Time-activity curves of whole brain, cortex, and cerebellum for baseline and displacement PET imaging studies of $[^{18}\text{F}]\text{OXD}-2314$ in the same monkey shown in Figure 3. The displacing dose of unlabeled OXD-2314 (0.5 mg/kg) was administered at 10 min post-injection (indicated by arrows).

The rate of uptake and clearance of $[^{18}\text{F}]\text{OXD}-2314$ in the brains of a male and a female baboon were compared (**Figure 5A**). Initial whole brain uptake was rapid in both baboons with peak uptake achieved in ~ 2 min with an SUV of ~ 2.8 in the male baboon and an SUV of ~ 1.9 in the female baboon. Rapid clearance of brain radioactivity was then observed in both baboons providing a 2:90 min SUV ratio of ~ 3.7 in the male baboon and ~ 3.6 in the female baboon. Radiometabolite analysis of $[^{18}\text{F}]\text{OXD}-2314$ in both baboons (**Figure 5B**) demonstrated a similar rate of $[^{18}\text{F}]\text{OXD}-2314$ metabolism over the course of the study. It is not known whether the higher initial brain uptake of $[^{18}\text{F}]\text{OXD}-2314$ observed in a male baboon is due to sex differences, normal physiologic variability, or the additional variability introduced by the effects of anesthesia. No evidence of elevated uptake or retention in the baboon striatum following $[^{18}\text{F}]\text{OXD}-2314$ injection was observed (**Figure S18**), a region where high levels of PSP and CBD pathology are expected.

Therefore, the normal brain retention patterns of [¹⁸F]OXD-2314 are unlikely to confound the detection of specific signal from PSP and CBD pathology in human subjects.

Figure 5. Time activity curves of [¹⁸F]OXD-2314 in male and female baboon brain (A) and the fraction of unchanged [¹⁸F]OXD-2314 in plasma from 2 to 90 min post-injection (B).

The brain uptake of [¹⁸F]OXD-2314 in rat and NHP's at early time points was above the desired threshold of 1 SUV, with a ratio between the initial maximum of brain radioactivity compared to late minimum brain radioactivity of ~8 in rats and ~4 in NHP (**Figures 2-5**). Early-to-late ratios of this magnitude in the brains of healthy animals lacking tau aggregates indicate suitably high brain penetration and rapid clearance properties desired of an efficacious brain imaging PET radiotracer."

The Images in the monkey and baboons show a relatively homogeneous late time distribution of radioactivity in cortical and subcortical areas, with striatum, globus pallidus, and brainstem values similar to cortex. A displacement study in a monkey with 0.5 mg/kg OXD-2314 was done that compares baseline and displacement TACs to evaluate the specific vs non-specific retention in brain. We have added a striatum ROI TAC for the baboon PET scans to SI as figure S18. As seen below:

Figure S18. Time activity curves in a male baboon following [^{18}F]OXD-2314 injection showing no preferential radiotracer retention in striatum.

3) in S1, deriving AUC for OXD-2115 with data from only 2 time points is very imprecise and should be avoided, particularly because the C_{max} is not clearly defined. Instead the brain to plasma ratio should be indicated at one or two selected time points for both compounds. Unbound fractions of both compounds in plasma and brain should also be indicated.

The data for OXD-2115 only contains 2 time points due to the poor permeability of the compound. For full transparency the complete results and protocols from the mouse pharmacology analysis has been added to SI pages 6-12.

4) in the method section for the radiobinding assays, the use of total brain homogenates or of a specific biochemical fractions should be clearly indicated.

The section has been amended to specify that only human brain homogenates were used:

“Postmortem tissues - Homogenates. Postmortem human brain tissues were provided by brain banks at the University of California, San Francisco (UCSF) and Banner/Sun Health AZ following informed consent of the donors and utilized at the University of Pittsburgh under the approval of the Committee for Oversight of Research and Clinical Training Involving Decedents. See Table S1 in Supporting Information for demographic and neuropathological characterizations of the postmortem tissues. The binding assays utilized homogenates of fresh frozen, autopsy-confirmed, postmortem human AD, PSP, CBD, PiD, and control brain tissue blocks (1 cm³) obtained from the Neurodegenerative Disease Brain Bank at UCSF, which contained only frequent mixed 3R/4R-tau neurofibrillary tangles and Ab plaque aggregates (AD tissue, middle frontal gyrus), or only 4R-tau aggregates (PSP tissue, superior frontal gyrus or putamen/globus pallidus, or CBD tissue, middle frontal gyrus or putamen), or only 3R-tau

aggregates (PiD tissue, middle frontal gyrus) and no other detectable aggregated amyloid species, or no detectable aggregated amyloid species at all (control tissue, middle frontal gyrus). Fresh frozen, autopsy-confirmed human PD anterior cingulate cortex brain tissue blocks (1 cm³) were obtained from Dr. Thomas Beach at Banner/Sun Health AZ with funding from the Michael J. Fox Foundation and contained frequent α -synuclein aggregates and no other detectable aggregated amyloid species. The frozen tissue blocks were prepared for the binding assays as previously described,^{46,47} except that equal quantities of the homogenates of three AD brains (AD1-AD3, Table S1) were pooled, as were homogenates from three PSP brains (PSP1-PSP3), three CBD brains (CBD1-CBD3), three Pick's brains (PiD1-PiD3), and 4 PD brains (PD1-PD4) in order to help minimize the inter-subject variability within the different neuropathological groups. Individual subcortical brain tissues from a PSP subject (PSP4) and a CBD subject (CBD4) as well as with the middle frontal cortex of a control subject were prepared for the binding assays as previously described."

5) additional cross-competition studies could be added to provide more information on the possible binding site of OXD-2314 (e.g. competition of OXD-2314 with MK-6240 in AD and with Florzolotau in PiD)

Additional cross competition studies have now been carried out according to the Reviewers suggestions. The revised Table 4 (formerly Table 3 as stated above) reveals competition of [³H]OXD-2314 against MK-6240, PI-2620, Tauvid, florzolotau, OXD-2115 and self blocking in AD, PSP, CBD and PiD tissue. This data has been added to Table 4 in the revised manuscript (*vide supra*).

Reviewer #2 (Remarks to the Author):

This study assessed the utility of a new potential tau PET ligand, [18F]OXD-2314, as a high affinity ligand for binding to 4-repeat (4R) tau. The authors identified OXD-2314 by assessing the in vitro tau affinity of over 150 analogues of OXD-2115. They showed that OXD-2314 had higher affinity to tau aggregates in Alzheimer's disease, progressive supranuclear palsy (PSP) and corticobasal degeneration (CBD) than three other ligands (PI-2620, florzolotau and OXD-2115) and demonstrated that OXD-2314 binds to a different high affinity site on tau aggregates compared to PI-2620 and florzolotau. They also found no off-target binding to MAO-A or MAO-B. Suitably high brain penetration and rapid clearance properties were identified in mice studies. The authors conclude that OXD-2314 is ready to be translated for first-in-human PET studies.

This work is very high impact with many laboratories and researchers wanting a tau PET ligand to investigate 4R tauopathies. The studies presented in this manuscript are essential steps to allow the use of OXD-2314 in humans. I do not have any major concerns, but have some comments/questions for the authors to address:

We thank the Reviewer for their enthusiastic support of our manuscript for publication and have addressed all of their insightful comments and questions raised.

1. The tissue sample for the PSP patients came from the superior frontal gyrus, rather than brainstem or subcortical regions which would be expected to have the greatest tau burden in a typical PSP case. Was the pathology in the PSP and CBD case/cases atypical? Was the clinical diagnosis of the PSP case/s PSP and for CBD was it CBS? That is, were these typical PSP and CBD cases?

The PSP and CBD cases were typical as was their neuropathology. A new Neuropathology Table added to supporting information as Table S1 is shown below and addresses both comments 1) and 2). Table S1 confirms that the PSP, CBD, and PiD tissues provided by UCSF did not have 3R+4R pathology.

Table S1. Summary of demographic and neuropathologic characteristics of donors and postmortem brain tissues. Key to scores: 1 = mild, 2 = moderate, 3 = severe ; or 1 = small, 2 = moderate, or 3 = large numbers of given hallmark.

Case	Sex	Age at death	Clinical Diagnosis	Neuropathology Diagnosis	PMI (hrs)	Frozen Block Anatomical Location	Abeta_CERAD_Neuritic plaques	Abeta_Diffuse Plaques	Abeta_CAA	CP13_Neurofibrillary tangles
AD1	M	63	AD	AD	6.0	Middle/inferior frontal gyrus, anterior	3	3	0	3
AD2	M	64	AD	AD	6.8	Middle/inferior frontal gyrus, anterior	3	3	1	3
AD3	F	66	AD	AD	15.4	Middle/inferior frontal gyrus, anterior	3	3	2	3
PSP1	M	63	PSP	PSP	4.4	Superior frontal gyrus	0	0	0	0
PSP2	F	70	PSP	PSP	7.2	Superior frontal gyrus	0	0	0	0
PSP3	M	68	PSP	PSP	10.1	Superior frontal gyrus	0	0	0	0
PSP4	F	66	PSP	PSP	9.2	Putamen & Globus Pallidus	0	0	0	0
CBD1	M	68	PNFA	CBD	7.4	Middle/inferior frontal gyrus, anterior	0	0	0	0
CBD2	F	70	CBS	CBD	7.4	Middle/inferior frontal gyrus, anterior	0	0	0	0
CBD3	F	63	CBS	CBD	5.0	Middle/inferior frontal gyrus, anterior	0	0	0	0
CBD4	M	75	CBS	CBD	7.4	Putamen	0	0	0	0
PID1	M	64	CBS	Pick's	13.7	Middle/inferior frontal gyrus, anterior	0	1	0	0
PID2	F	63	PNFA	Pick's	4.3	Middle frontal gyrus, anterior	0	0	0	0
PID3	F	69	bvFTD	Pick's	6.3	Middle/inferior frontal gyrus, anterior	0	0	0	0
PD1	M	67	PD	PD	4.2	Anterior cingulate	0	0	0	0
PD2	F	72	PD	PD	6.1	Anterior cingulate	0	0	0	0
PD3	F	73	PD	PD	7.3	Anterior cingulate	0	0	0	0
PD4	M	69	PD	PD	5.3	Anterior cingulate	0	0	0	0
Control	F	86	Control	Control	6.4	Middle/inferior frontal gyrus, anterior	0	0	0	0

Case	CP13_Pick bodies	CP13_Neuronal cytoplasmic inclusions	CP13_Globose tangles	CP13_Astrocytic plaques	CP13_Tufted astrocytes	CP13_Thorny astrocytes	CP13_Neuropil threads/grains	CP13_White matter threads/grains	CP13_Glial cytoplasmic inclusions	aSyn_Lewy Bodies	aSyn_Lewy Neurites	aSyn_Glial cytoplasmic inclusions
AD1	0	0	0	0	0	0	3	1	0	0	0	0
AD2	0	0	0	0	0	0	3	1	0	0	0	0
AD3	0	0	0	0	0	0	3	1	0	0	0	0
PSP1	0	3	0	0	3	1	2	2	3	0	0	0
PSP2	0	2	0	0	3	1	1	1	3	0	0	0
PSP3	0	2	1	0	3	1	1	0	2	0	0	0
PSP4	0	2	1	0	3	0	1	1	3	0	0	0
CBD1	0	3	0	3	0	0	3	3	2	0	0	0
CBD2	0	3	0	3	0	1	3	3	3	0	0	0
CBD3	0	2	0	3	0	0	2	2	1	0	0	0
CBD4	0	3	0	3	0	0	3	3	3	0	0	0
PID1	3	1	0	0	0	0	3	1	3	0	0	0
PID2	3	1	0	0	0	0	2	2	3	0	0	0
PID3	3	1	0	0	0	0	3	2	3	0	0	0
PD1	0	0	0	0	0	0	0	0	0	3	3	0
PD2	0	0	0	0	0	0	0	0	0	2	3	0
PD3	0	0	0	0	0	0	0	0	0	3	3	0
PD4	0	0	0	0	0	0	0	0	0	3	3	0
Control	0	0	0	0	0	0	0	0	0	0	0	0

Additional K_d and B_{max} values for the subcortical regions in PSP and CBD were acquired and has been added to Tables 2 and 3 in the revised manuscript as shown below:

Table 2. *In vitro* equilibrium dissociation constant (K_d) values ($n = 3$; mean \pm SD) of [^3H]PI-2620, [^3H]florzolotau, [^3H]OXD-2115, and [^3H]OXD-2314 in AD, PSP, CBD, PiD, and PD frontal cortex tissues (see Table S1 for demographics and neuropathology of subjects). (n.d. = not determined; undefined = insufficient specific binding to fit the data). The AD, PSP, CBD, and PiD homogenates were pooled from 3 cases, the PD homogenates were pooled from 4 cases, and the Control was a representative single subject. * Subcortical tissue

Radiotracer	AD K_d (nM)	PSP K_d (nM)	CBD K_d (nM)	PiD K_d (nM)	PD K_d (nM)	Control K_d (nM)
[^3H]PI-2620	2.5 \pm 1.8	5.3 \pm 4.6	23 \pm 4	n.d.	n.d.	undefined
[^3H]Florzolotau	4.1 \pm 0.5	4.6 \pm 1.0	36 \pm 7	20 \pm 3	5.1 \pm 1.1	undefined
[^3H]OXD-2115	5.5 \pm 1.9	4.9 \pm 0.2	27 \pm 1	34 \pm 12	48 \pm 11	undefined
[^3H]OXD-2314	3.6 \pm 0.7	2.4 \pm 0.6 2.3 \pm 0.3*	2.1 \pm 0.3 2.2 \pm 0.4*	1.1 \pm 0.2	62 \pm 8	undefined

Table 3. B_{\max} values for [^3H]OXD-2314 in AD, PSP, PiD and PD tissue.

Radiotracer	Brain region	AD ($n = 3$) B_{\max} (nM)	PSP ($n = 3$) B_{\max} (nM)	CBD ($n = 3$) B_{\max} (nM)	PiD ($n = 3$) B_{\max} (nM)	PD ($n = 1$) B_{\max} (nM)
[^3H]OXD-2314	Frontal cortex	430 \pm 110	130 \pm 50	96 \pm 32	63 \pm 21	240
	Subcortical	-	170 \pm 40	110 \pm 30		

2. Very little information is provided on the 4R tau cases used in this study. How many PSP, CBD, PiD and AD cases were included? What was the Braak NFT stage of the PSP, CBD and PiD cases. This is important information to determine whether the 4R tau cases had co-existing 3R+4R tau pathology that the ligand could be binding.

The requested information on the 4R tau cases used in the studies are now addressed in response to Question 1 (*vide supra*).

3. Radioactivity washout was slower in the female rats (with higher overall brain radioactivity) compared to the male rats – could the authors discuss this finding and whether this may have implications for sex differences in humans.

The sex differences observed in rats are indeed noteworthy. Both kinetics and metabolism are different. Additional PET scans in male and female baboons were carried out and this data has been included in the revised manuscript. Similar metabolic rates were seen between male and female baboons with some differences in early brain uptake. The exact cause of this effect is

beyond the scope of this study, but will be further investigated in our upcoming human PET studies with [¹⁸F]OXD-2314.

The revised section of the manuscripts reads as below.

“Dynamic PET imaging was performed in a rhesus macaque (*macaca mulatta*) monkey following the injection of [¹⁸F]OXD-2314 at baseline (104.9 MBq, *n* = 1) and was repeated in a second [¹⁸F]OXD-2314 study (115.8 MBq, *n* = 1) where a displacing dose of unlabeled OXD-2314 (0.5 mg/kg) was administered 10 min after [¹⁸F]OXD-2314 injection to verify the absence of specific binding in NHP brain. Venous blood samples were drawn to characterize the metabolic fate of [¹⁸F]OXD-2314 in the macaque. Dynamic [¹⁸F]OXD-2314 PET imaging was also performed in a male (106.6 MBq, *n* = 1) and a female (115.8 MBq, *n* = 1) baboon (*papio anubis*), and venous blood samples were drawn in both baboons to characterize potential effects of sex on radiometabolism. Bilateral regions of interest were defined on 5-8 contiguous axial image planes for the lateral temporal cerebral cortex, cerebellum, striatum, and whole brain, which were used to sample dynamic [¹⁸F]OXD-2314 PET images to determine the time-varying radioactivity concentration expressed in SUV units. As shown in Figure 3A, [¹⁸F]OXD-2314 readily entered macaque brain (whole brain peak ~2.2 SUV at ~2 min) and distributed uniformly throughout cortical and subcortical grey matter regions. Radioactivity cleared rapidly from the whole brain with a clearance half-time of ~22 min and resulted in a 2:90 min radioactivity concentration ratio of 3.9 (Figure 3B). No preferential retention of [¹⁸F]OXD-2314 in any brain region was observed, suggesting that signal in NHP brain was dominated by non-displaceable radioactivity (non-specifically bound and free radiotracer). Administration of a pharmacologic dose of unlabeled OXD-2314 (0.5 mg/kg) 10 min after [¹⁸F]OXD-2314 injection (Figure 4) resulted in a very small transient increase in brain radioactivity concentrations, possibly resulting from an increase in plasma concentrations of free [¹⁸F]OXD-2314 released from peripheral sites. There was no evidence of accelerated brain clearance in any brain region that would suggest a significant component of displaceable (specifically bound) [¹⁸F]OXD-2314 to any site in normal monkey brain. These data are consistent in showing that non-displaceable binding is the only significant source of [¹⁸F]OXD-2314 signal in normal NHP brain. Radiometabolite analysis of macaque venous blood identified only one, polar radiometabolite, which increased from 37% at 10 min to 77% at 90 min post injection and was not likely brain penetrant (**Figures S19-S20**).

Figure 3. A) PET summation images (0-11 min and 60-90 min) and B) time-activity curves of whole brain (●), cortex (▲), and cerebellum (○) from NHP PET imaging of $[^{18}\text{F}]\text{OXD}-2314$.

Figure 4. Time-activity curves of whole brain, cortex, and cerebellum for baseline and displacement PET imaging studies of $[^{18}\text{F}]\text{OXD}-2314$ in the same monkey shown in Figure 3. The displacing dose of unlabeled OXD-2314 (0.5 mg/kg) was administered at 10 min post-injection (indicated by arrows).

The rate of uptake and clearance of $[^{18}\text{F}]\text{OXD}-2314$ in the brains of a male and a female baboon were compared (Figure 5A). Initial whole brain uptake was rapid in both baboons with peak uptake achieved in ~ 2 min with an SUV of ~ 2.8 in the male baboon and an SUV of ~ 1.9 in the female baboon. Rapid clearance of brain radioactivity was then observed in both baboons providing a 2:90 min SUV ratio of ~ 3.7 in the male baboon and ~ 3.6 in the female baboon. Radiometabolite analysis of $[^{18}\text{F}]\text{OXD}-2314$ in both baboons (Figure 5B) demonstrated a similar rate of $[^{18}\text{F}]\text{OXD}-2314$ metabolism over the course of the study. It is not known whether the higher initial brain uptake of $[^{18}\text{F}]\text{OXD}-2314$ observed in a male baboon is due to sex differences, normal physiologic variability, or the additional variability introduced by the effects of anesthesia. No evidence of elevated uptake or retention in the baboon striatum following $[^{18}\text{F}]\text{OXD}-2314$ injection was observed (**Figure S18**), a region where high levels of PSP and CBD pathology are expected.

Therefore, the normal brain retention patterns of [¹⁸F]OXD-2314 are unlikely to confound the detection of specific signal from PSP and CBD pathology in human subjects. ”

Figure 5. Time activity curves of [¹⁸F]OXD-2314 in male and female baboon brain (A) and the fraction of unchanged [¹⁸F]OXD-2314 in plasma from 2 to 90 min post-injection (B).

4. It is very useful that the authors assessed binding to MAO-A and MAO-B, but it would be very important to also assess binding to other non-tau targets that are a problem in 4R tauopathies, such as iron in the basal ganglia and neuromelanin in the substantia nigra.

Additional experiments to assess [³H]OXD-2314 binding to MAO-A and MAO-B was carried out in competition studies against deprenyl and R041-1049. Neither of which can effectively block [³H]OXD-2314 binding in AD, PSP or CBD tissue. This data has been added in Table S4 as seen below. Binding to iron and neuromelanin has been reported primarily for [¹⁸F]Tauvid (for example: Coakeley, S. *et al. Brain Struct Funct* **223**, (2018) and Choi, Jae Yong, *et al. J Nucl Med* 59.1 (2018). To our knowledge there is no established method for assessing this off-target binding *in vitro*. Investigation of off-target binding beyond the CNS targets assessed in the Eurofins screen will be looked at further in human PET scans, if warranted.

Table S4. Blocking study using [³H]OXD-2314 in AD, PSP and CBD tissue against MAO-A and MAO-B substrates deprenyl and Ro41-1049.

Radioligand	Cold Compound	Tissue	K _i (nM)
[³ H]OXD-2314	Deprenyl	AD (n = 1)	2900
[³ H]OXD-2314	Deprenyl	PSP (n = 1)	3000
[³ H]OXD-2314	Deprenyl	CBD (n = 1)	5000

[³ H]OXD-2314	Ro41-1049	AD (n = 1)	>10,000
[³ H]OXD-2314	Ro41-1049	PSP (n = 1)	>10,000
[³ H]OXD-2314	Ro41-1049	CBD (n = 1)	>10,000

5. It would be useful for the authors to show a picture of the uptake in a PSP and or CBD case compared to the uptake in an AD case, even if in the superior frontal lobe. This is to get a sense of whether the signal is PSP (or CBD) is truly any different from what we are seeing with flortaucipir and the other ligands and how different it is from what occurs in AD.

In light of the Reviewer's comment, a new batch of [³H]OXD-2314 was synthesized and autoradiography was performed in AD and PSP tissue and compared to healthy controls and tau immunohistochemistry. Additional figures and text in the manuscript has been added and refers to the revised Supporting Information (Figures S21-S22 and Table S5) as follows:

"Autoradiography studies were also conducted with [³H]OXD-2314 in healthy control, AD, CBD and PSP tissues for total radiotracer signal and blocking with unlabeled OXD-2314, and compared to AT8 immunohistochemistry for tau staining. Signal was displaceable under homologous blocking conditions to the level of nonspecific binding present in healthy control tissue and elevated radiotracer signal was observed in AD and PSP hippocampal tissues compared to control."

Autoradiography using [³H]OXD-2314

Figure S21. [³H]OXD-2314 binding in HC, AD and PSP hippocampal tissue. Total [³H]OXD-2314 (3 nM) binding is shown along with displacement by unlabeled OXD-2314 (10 μM) compared to AT8 immunostaining for tau. Increased radiotracer signal was observed in AD and PSP tissue compared to healthy control. Radiotracer signal in AD and PSP tissue was displaced by homologous blockade and was highest in the AD cases.

Figure S22. [³H]OXD-2314 binding in CBD and PSP cortical tissue. Total [³H]OXD-2314 (3 nM) binding is shown along with displacement by unlabeled OXD-2314 (3 μM) compared to AT8 immunostaining for tau.

Table S5. Tissue Demographics used in autoradiography with [³H]OXD-2314.

Primary Pathological Diagnosis	Case ID	Region	Sex	Age
HC; A0B1C0; Low NFT	HC1	BA28/HP	F	72
Healthy	HC2	BA28/HP	F	51
AD; A3B2C3	AD1	BA28/HP	M	90
AD; A3B3C3	AD2	BA28/HP	M	88
AD; A3B3C3	AD3	BA28/HP	F	93
AD; Braak IV/V	AD4	BA28/HP	F	96
PSP	PSP1	BA28/HP	M	65
PSP	PSP2	BA28/HP	F	72

*BA: Brodmann's area; M: male; F: female

Reviewer #3 (Remarks to the Author):

The manuscript by Lindberg et al. describes their efforts to develop a 4R-tau PET tracer. The authors have previously reported a lead, [18F]OXD-2115 that showed good affinity toward 4R, but limited brain uptake. Their submitted manuscript describes the work toward optimizing this previous lead compound with regard to brain permeability. Around 150 compounds were explored and screened for tau affinity, and from these [18F]OXD-2314 was selected as the most promising candidate for further evaluation. The topic is timely, highly relevant and nicely introduced, describing the importance of a tracer for non-AD tauopathies. However, there are some aspects of the work for which further information could be helpful for the readers. One major comment; the aim was to develop a ligand selective toward 4R-tau, but the selected candidate binds with high affinity to both 3R and 4R-tau. I believe this has to be to be stronger acknowledged in the manuscript. Also, the title of the manuscript indicates a selective 4R-tau PET tracer. Nevertheless, the tracer has high affinity for non-AD tauopathies and the presented work can stimulate further advances in the field. Below are some of the concerns and question I have on the current version of the manuscript.

We thank the Reviewer for their support of our manuscript. We agree with the Reviewer regarding the distinction of [18F]OXD-2314 as a 4R-tau PET radiotracer and we have updated the manuscript to reflect that point and revised the title so that the emphasis is now on non-Alzheimer's disease tauopathies as follows:

Title is revised as follows:

"Ligand-based design of pyridinyl-indole PET radiotracers for non-Alzheimer's disease tauopathies: Development of [18F]OXD-2314 for human use"

Abstract is revised as follows:

"To date, no high-affinity non-AD tau- PET radiopharmaceutical has been optimized for imaging non-AD tauopathies."

Introduction page 5 now includes the following text:

"In the present study, novel pyridinyl-indole analogues of OXD-2115 were identified using ligand-based design to select for high potency and selectivity for non-AD tau aggregates as well as improved brain uptake.

Results, *In vitro* binding assay section is revised as follows:

"Overall, [3H]OXD-2314 was shown to be a high affinity ligand for tau aggregates in both AD and non-AD tauopathies.";

and:

"It is noteworthy that [3H]OXD-2115 showed high affinity to the primarily 4R-tau (PSP, CBD) aggregates in vitro but had lower affinity to the predominantly 3R-tau aggregates in PiD, and [3H]OXD-2314 was expected to be similar, but instead showed high affinity for all of the non-AD and mixed 3R/4R-tauopathies (AD) in human brain homogenate affinity assays (Table 2)."

Discussion is revised as follows:

"These results indicate that [³H]OXD-2314 is a selective high-affinity tau PET radiotracer with unique binding sites on tau aggregates for non-AD tauopathies."

Major comments

1) The title of the manuscript "Structure-guided design of pyridinyl-indole 4R-tau PET radiotracers: Development of [¹⁸F]OXD-2314 for human use" implies;
- that the authors made use of a protein structure/binding mode of the lead during the design phase. The term is also used in the manuscript (page 5 line 77). Consider using the term ligand-based design instead.

We thank the Reviewer for this suggestion and we have amended the manuscript to include the term "ligand-based design" throughout the manuscript as follows:

Title is revised as follows:

"Ligand-based design of pyridinyl-indole PET radiotracers for non-Alzheimer's disease tauopathies: Development of [¹⁸F]OXD-2314 for human use"

Introduction page 5 now includes the following text:

"In the present study, novel pyridinyl-indole analogues of OXD-2115 were identified using ligand-based design to select for high potency and selectivity for non-AD tau aggregates as well as improved brain uptake.

-that the tracer is selective toward 4R-tau over 3R-tau. However, the presented data shows that the tracer has a high affinity for 3R-tau as well. The authors also state this on page 12 lines 201-209. Consider changing "4R-tau" to "3R/4R-tau" or "tracer for non-AD tauopathies". The same is with the abstract line 21.

We thank the Reviewer for pointing this out. We agree with this characterization and changes have been made to include "non-AD tau" rather than focusing on a specific isoform, as seen in the first response to the Reviewer, and restated below:

Abstract:

"To date, no high-affinity non-AD tau- PET radiopharmaceutical has been optimized for imaging non-AD tauopathies."

Introduction page 5:

"In the present study, novel pyridinyl-indole analogues of OXD-2115 were identified using ligand-based design to select for high potency and selectivity for non-AD tau aggregates as well as improved brain uptake.

Results, *In vitro* binding assay:

"Overall, [³H]OXD-2314 was shown to be a high affinity ligand for tau aggregates in both AD and non-AD tauopathies."

"It is noteworthy that [3H]OXD-2115 showed high affinity to the primarily 4R-tau (PSP, CBD) aggregates in vitro but had lower affinity to the predominantly 3R-tau aggregates in PiD, and

[3H]OXD-2314 was expected to be similar, but instead showed high affinity for all of the non-AD and mixed 3R/4R-tauopathies (AD) in human brain homogenate affinity assays (Table 2)."

Discussion:

"These results indicate that [3H]OXD-2314 is a selective high-affinity tau PET radiotracer with unique binding sites on tau aggregates for non-AD tauopathies."

2) Page 5 line 79: "Lead candidates were further evaluated against [3H]OXD-2115 in human postmortem AD, CBD and PSP tissues for comparison with first- and second-generation PET radiopharmaceuticals for imaging tau aggregation."

What candidates were tested on postmortem tissues? Only data for the selected tracer candidate is shown in table 2 and 3.

We apologize for the confusion and we have reclarified the caption in the Table 1. All 6 of the candidate compounds as well as standard OXD-2115 in Table 1 were evaluated on postmortem tissues and assayed against [³H]OXD-2115 in AD and PSP tissue. The reference to assays in CBD tissue for these compounds has been deleted in the revised manuscript. For clarity the Table 1 caption now reads as follows:

Table 1. Inhibition constant (K_i) values (n=1) of OXD-2115 and analogues vs. [³H]OXD-2115 in AD and PSP tissue homogenates to establish inhibition potencies. *In silico* models for predicting BBB permeability included CNS MPO, CNS PET MPO and BBB scores. Chemdraw Professional 15.0 was used for all physicochemical properties except logD_{7.4} which was obtained experimentally.

Extensive binding data for all compounds are available in the patents cited in the manuscript:

Sohn, D. WO 2019/197502 A1; Selective ligands for tau aggregates. (2019).

Sohn, D. WO 2021/074351 A1; Preparation of benzothiazoles/ indoles as selective ligands for tau aggregates. United Kingdom patent (2021).

3) The authors synthesized more than 150 compounds with R-group exploration in two different positions, with the aim to improve the physicochemical properties for brain uptake. It is not clear from the text if they made use of the *in silico* prediction scores (BBB and CNS (PET) MPO scores) already at this stage, or if scoring was only performed for the few selected leads presented in table 1?

Table 1 has been extended to include *in silico* scores and clogP have for all the candidates along with discussion for the selection criteria, as follows:

"The BBB score was therefore considered for identifying the next lead. Primarily due to reducing the number of HBDs, four leads showed improved BBB scores compared to OXD-2115 (OXD-2317, OXD-2312, OXD-2313 and OXD-2314; see Table 1) and were equipotent in AD and PSP tissue. However, OXD-2317, OXD-2312 and OXD-2313 had higher lipophilicity (clogP), which may lead to higher non-specific binding and/or plasma protein binding *in vivo*. OXD-2314 was identified as the most promising high-affinity compound for PSP-tau with potential of BBB permeability and was carried forward for further evaluation (Table 1)."

Table 1. Inhibition constant (K_i) values (n=1) of OXD-2115 and analogues vs. [^3H]OXD-2115 in AD and PSP tissue homogenates to establish inhibition potencies. *In silico* models for predicting BBB permeability included CNS MPO, CNS PET MPO and BBB scores. Chemdraw Professional 15.0 was used for all physicochemical properties except $\log D_{7.4}$ which was obtained experimentally.

Compound	AD tissue K_i (nM)	PSP tissue K_i (nM)	ClogP	CNS MPO	CNS PET MPO	BBB score
 OXD-2115	20	27	2.35	3.7	1.9	3.18
 OXD-2310	21	6.6	1.74	4.2	3.0	3.16
 OXD-2331	7.4	6.3	2.48	4.1	1.9	3.17
 OXD-2317	5.4	5.8	3.52	3.7	1.8	4.07
 OXD-2312	12	5.1	3.66	3.6	1.7	4.07
 OXD-2313	11	4.8	3.66	3.6	1.7	4.07
 OXD-2314	8.8	4.8	3.25	3.8	1.7	3.65

Why not include the structures/affinity and in silico predictions scores for these ~150 compounds as a table in the supporting information? The SAR can also be summarized in a figure instead of the current scheme 1, along with (or alternatively) graphs displaying any trends in properties (e.g., BBB and MPO scores, HBDs, lipophilicity, TPSA, etc) vs. affinity. This is extensive work and an important part of the project, but it's given too little attention in the current state of the manuscript.

We appreciate the Reviewers interest in the full 150 compounds synthesized. All compounds and their affinities are covered in depth in the patents cited, and we have revised the text as follows:

"Synthesis of structural analogs of OXD-2115 focused on diversification of either the R1 substituent on the indole hydroxyl group or the R2 substituent on the central pyridinyl ring (Scheme 1).³⁴"

Sohn, D. WO 2019/197502 A1; Selective ligands for tau aggregates. (2019).

Sohn, D. WO 2021/074351 A1; Preparation of benzothiazoles/ indoles as selective ligands for tau aggregates. United Kingdom patent (2021)

On a small note, the first screening of the ~150 compounds, was this displacement at a single point concentration or directly full binding curves for Ki determination?

In the first screening a single point concentration was used. This is clarified in the text as follows:

"A library of over 150 novel compounds were synthesized based on OXD-2115 and screened against [³H]OXD-2115 in AD and PSP tissues. Compounds with Ki values below 200 nM were selected for further screening."

4) Page 6 line 100-101: "Two enantiomeric analogues of OXD-2115, namely, OXD-2188 and OXD-2189, were initially identified as promising candidates."

Are these the two enantiomers of the initial lead compound OXD-2115 or enantiomers of a structural analog of OXD-2115? I assume it is the latter since on page 9 line 147-148, the authors write that these two tracers had lower lipophilicity than OXD-2115, which would not be the case if they are enantiomers. Please clarify this in the first sentence, and add the structures to the main manuscript or in the supporting information (combined with the presented PET data). In addition, assuming these compounds were enantiomers of a structural analog of OXD-2115: What was the affinity in vitro and was it different between the enantiomers? Even though they have similar in vitro affinity the PK, ADME(T), etc can of course differ, which justifies evaluating both in vivo. However, since the structures are not specified it does not add much to the manuscript. Either add more information regarding these compounds, including structures or consider to take this out.

The first sentence has been clarified in text as follows:

"Two analogues of OXD-2115, the enantiomeric pair, OXD-2188 and OXD-2189, were initially identified as promising candidates."

To address the second point regarding the in vitro data for OXD-2188 and OXD-2189. The in vitro evaluation and structures has been added to SI page 3 as follows:

"In vitro characterization. K_i values for OXD-21188 and OXD-2189 vs. [3 H]OXD-2115 in PSP tissue was 29.8 nM and 13.7 nM respectively."

5) It is not totally clear why OXD-2314 was selected as the most promising lead. The K_i values presented in table 1 are from $n=1$, which is fine for a preliminary screening. However, the last 5 compounds in table 1 have very similar profile. Did the in silico prediction scores of these compounds differ a lot? Were those scores used to prioritize the compounds? Adding columns with the scores for each compound to the table would be helpful. In addition, a sentence about your criteria and thresholds for affinity in the different tissue homogenates would be good for clarification. Do you want equally potent and/or higher in AD tissue?

The in silico scores for all compounds have been included in Table 1. CNS MPO scores vary from 3.6 -4.2; CNS PET MPO between 1.7-3.0; and BBB score between 3.16 and 4.07. The scores were indeed used to prioritize the compounds. The rationale for selection of OXD-2314 has been further explained. Regarding the thresholds for affinity in different tissue homogenates, the manuscript has now been rewritten to describe the lead compounds as potent and selective towards AD and non-AD tauopathies, and we have clarified that the lead compounds in this study are equipotent to AD and PSP tissues. All of these points have been clarified in the revised manuscript as follows:

"The BBB score was therefore considered for identifying the next lead. Primarily due to reducing the number of HBDs, four leads showed improved BBB scores compared to OXD-2115 (OXD-2317, OXD-2312, OXD-2313 and OXD-2314; see Table 1) and were equipotent in AD and PSP tissue. However, OXD-2317, OXD-2312 and OXD-2313 had higher lipophilicity (clogP), which may lead to higher non-specific binding and/or plasma protein binding in vivo. OXD-2314 was identified as the most promising high-affinity compound for PSP-tau with potential of BBB permeability and was carried forward for further evaluation (Table 1)."

Per the Reviewer's suggestion, Table 1 has also been extended with additional columns to include CNS MPO, CNS PET MPO, BBB score and clogP values used for the selection process as follows:

Table 1. Inhibition constant (K_i) values ($n=1$) of OXD-2115 and analogues vs. [3 H]OXD-2115 in AD and PSP tissue homogenates to establish inhibition potencies. *In silico* models for predicting BBB permeability included CNS MPO, CNS PET MPO and BBB scores.

Compound	AD tissue K_i (nM)	PSP tissue K_i (nM)	ClogP	CNS MPO	CNS PET MPO	BBB score
 OXD-2115	20	27	2.35	3.7	1.9	3.18

 OXD-2310	21	6.6	1.74	4.2	3.0	3.16
 OXD-2331	7.4	6.3	2.48	4.1	1.9	3.17
 OXD-2317	5.4	5.8	3.52	3.7	1.8	4.07
 OXD-2312	12	5.1	3.66	3.6	1.7	4.07
 OXD-2313	11	4.8	3.66	3.6	1.7	4.07
 OXD-2314	8.8	4.8	3.25	3.8	1.7	3.65

6) The section called "BBB-permeability": See earlier comment if this section can be combined with the lead identification. An extensive description and discussion of the in silico prediction scores are given. These are tools to assist in the design and/or to prioritize compounds. As it is now, too much attention is given to these. This text can be shorten. More importantly, the threshold stated for high probability of brain permeability in each model is incorrect (Page 9 line 140-142). The threshold in each model for high likelihood of success is as follows: >4 (CNS MPO), >3 (CNS PET MPO), and >4 (BBB score). Meaning your candidate is slightly below the threshold in all scores.

Please keep in mind when discussing your result (page 9-10, line 153-158 + discussion) that these scores do not predict the same. The BBB score only predict passive permeability, so a

score above 4 means that a compound most likely cross the BBB. Whereas, the CNS MPO scores not only take into account permeability, but also ADME(T) properties (for tracers e.g., non-specific binding, p-gp liability), meaning you predict the likelihood of overall success as a CNS drug and PET tracer, respectively.

The two sections have been combined and shortened as follows:

“BBB permeability.

There are several computational approaches for obtaining high BBB permeability of central nervous system (CNS) PET radiotracers.³⁴ In addition to the preliminary assays, *in silico* calculations were made to predict the BBB permeability and lipophilicity of each lead compound. ClogP was first used to rank the leads based on lipophilicity. The use of computational *in silico* models offers a fast and high throughput screening method for identifying physicochemical contributors that might affect the BBB permeability of a radiotracer.³⁵ Three computational models were employed to evaluate potential improvements to the low brain uptake of OXD-2115: CNS MPO, CNS PET MPO, and BBB score.³⁶⁻³⁸ OXD-2115 scored 3.7/6.0 in CNS MPO, 1.9/6.0 in CNS PET MPO, and 3.18/6.0 in BBB score, which are all within the range for potentially CNS active compounds in each model (Figure S2). Evaluating the subscores for physicochemical properties of OXD-2115, the number of HBDs, the pKa of the indole nitrogen,³⁹ and molecular size are likely physicochemical contributors to the low brain uptake seen in PET scans using [¹⁸F]OXD-2115.³⁰ CNS MPO and CNS PET MPO also predicted that reducing the lipophilicity would increase BBB permeability. However, two radiotracers were explored with lower lipophilicities, [¹⁸F]OXD-2188 and [¹⁸F]OXD-2189, and neither showed improved brain uptake compared with [¹⁸F]OXD-2115 in rats. The BBB score was therefore considered for identifying the next lead. Primarily due to reducing the number of HBDs, four leads showed improved BBB scores compared to OXD-2115 (OXD-2317, OXD-2312, OXD-2313 and OXD-2314; see Table 1) and were equipotent in AD and PSP tissue. However, OXD-2317, OXD-2312 and OXD-2313 had higher lipophilicity (clogP), which may lead to higher non-specific binding and/or plasma protein binding *in vivo*. OXD-2314 was identified as the most promising high-affinity compound for PSP-tau with potential of BBB permeability and was carried forward for further evaluation (Table 1).”

Regarding the Reviewer’s comments on the use of the *in silico* models for the development of novel radiopharmaceuticals is addressed as follows: In this work we used the different models as part of forming an internal ranking between the potential candidates. While each model has certain cut-offs for likelihood for success, those thresholds (BBB score >4 for example) are by no means absolute, as shown in this case but is also shown in the primary literature to report each model, and therefore a combination of models were used in this case to predict optimal brain uptake for PET tracers in this structural class.

7) Information about the software used for calculating the properties has to be provided. The final score will differ depending on the software used for calculating the properties, especially for ClogP, pKa and ClogD7.4. For example, the CNS (PET) MPO is trained on descriptors from ChemDraw, Biolum and ACD Percepta. ChemAxon and MOE work fine with the BBB score.

If the authors have used ChemAxon, which has rather good correlation with the original software for CNS MPO prediction, I refer the authors to a paper from 2021 by Urbina et al. (ACS Chem. Neurosci. 2021, 12, 12, 2247–2253).

The above-mentioned comments have all been clarified in the Legends to Table 1 and Figure S2 as follows:

“**Table 1.** Inhibition constant (K_i) values (n=1) of OXD-2115 and analogues vs. [³H]OXD-2115 in AD and PSP tissue homogenates to establish inhibition potencies. In silico models for predicting BBB permeability included CNS MPO, CNS PET MPO and BBB scores. Chemdraw Professional 15.0 was used for all physicochemical properties except logD_{7.4} which was obtained experimentally.”

“**Figure S2.** In silico evaluation of CBD-2115 and OXD-2314 using CNS MPO, CNS PET MPO and BBB score. Chemdraw Professional 15.0 was used for all physicochemical properties except logD_{7.4} which was obtained experimentally.”

8) Not enough details are provided regarding the in vitro binding experiments. The authors refer to references 43 and 44 in the manuscript. I assume the assays have been performed similarly, but then this should be stated and the concentration ranges for each compound, amount of the different homogenates, conditions, etc for the assays in the present work should be added. Maybe the authors can also refer to their previous paper in ACS Chem Neurosci from 2021?

During submission of this manuscript, our newest publication with our fully detailed in vitro binding methods was published in the (Mathis et al. *In Silico Discovery and Subsequent Characterization of Potent 4R-Tauopathy Positron Emission Tomography Radiotracers. Journal of Medicinal Chemistry* **66**, 10628-10638 (2023) on the development of an ultrahigh throughput screening of 4R tau PET radiotracers based on an Enamine library screen, and has been referenced (reference 47), with the following text added:

Equilibrium dissociation constant (K_d) assays. Tritium-labeled radioligand binding assays utilized AD, PSP, CBD, PiD, PD, or PD brain homogenates to determine equilibrium dissociation constant (K_d) values and were performed as previously described.^{46,47}

In vitro competition (K_i) assays. The equilibrium inhibition constant (K_i) values of the unlabeled compounds were determined versus tritium-labeled radioligands using published methods.⁴⁶

In addition, how is the non-specific binding assessed in the experiments determining K_d of the 3H-ligands on different tissues? Self-blocking?

The assessment of non-specific binding was done by self-blocking with 1-3 μM of unlabeled OXD-2314. For full details we refer to a recent paper from our laboratories (Mathis et al. *In Silico Discovery and Subsequent Characterization of Potent 4R-Tauopathy Positron Emission Tomography Radiotracers. Journal of Medicinal Chemistry* **66**, 10628-10638 (2023)).

9) No experimental details are provided for the PK studies in mice.

Full details for the PK studies in mice have now been included in SI pages 6-12

10) Page 13, line 221-224 "Of note, OXD-2314 was found to inhibit the binding of the antagonist radioligand (protriptyline) to the norepinephrine transporter(h) by 62.9% and the enzyme activity of 5-lipoxygenase by 50."

How is the target abundance and regional expression levels of these off-targets compared to tau in the disease specific conditions the authors want to assess? If known, this information is worth adding, as well as if the authors foresee any potential issues here?

The expression of the two targets in questions were considered and the manuscript as been amended as follows:

"Norepinephrine transports are expressed in low concentration throughout the brain and its relative concentration are not affected by aging or neurodegeneration.⁴¹ 5-lipoxygenase may be overexpressed in aging brain and play some role in neurodegeneration, which would be necessary to further investigate in vivo.⁴² "

The new references are listed below:

41 Mandela, P. & Ordway, G. A. The norepinephrine transporter and its regulation. *Journal of Neurochemistry* 97, 310-333 (2006). <https://doi.org/10.1111/j.1471-4159.2006.03717.x>

42 Manev, H., Uz, T., Sugaya, K. & Qu, T. Y. Putative role of neuronal 5-lipoxygenase in an aging brain. *Faseb Journal* 14, 1464-1469 (2000). <https://doi.org/10.1096/fj.14.10.1464>

11) Why haven't the authors performed any studies in an animal model with tau filaments? For example, the 4R-tau expressing P301L mice?

We had not considered animal models with tau filaments for in vivo PET studies as the majority of tau-PET radiopharmaceuticals that have been translated for first in human use were not successful in these models and therefore cannot be relied upon for a go-no-go decision for first in human use. However, in light of the Reviewer's comment, we have now included in vitro binding assays of brain homogenates from the P301L mouse model in Table 2:

Table 2. *In vitro* equilibrium dissociation constant (K_d) values ($n = 3$; mean \pm SD) of [³H]PI-2620, [³H]florzolotau, [³H]OXD-2115, and [³H]OXD-2314 in AD, PSP, CBD, PiD, and PD frontal cortex tissues as well as P301L transgenic mouse tissue (see Table S1 for demographics and neuropathology of subjects). (n.d. = not determined; undefined = insufficient specific binding to fit the data). The AD, PSP, CBD, and PiD homogenates were pooled from 3 cases, the PD homogenates were pooled from 4 cases, and the Control was a representative single subject. * Subcortical tissue

Radiotracer	AD K_d (nM)	PSP K_d (nM)	CBD K_d (nM)	PiD K_d (nM)	PD K_d (nM)	P301L transgenic mice K_d (nM)	Control K_d (nM)
[³ H]PI-2620	2.5 \pm 1.8	5.3 \pm 4.6	23 \pm 4	n.d.	n.d.	n.d.	undefined
[³ H]Florzolotau	4.1 \pm 0.5	4.6 \pm 1.0	36 \pm 7	20 \pm 3	5.1 \pm 1.1	n.d.	undefined

[³H]OXD-2115	5.5 ± 1.9	4.9 ± 0.2	27 ± 1	34 ± 12	48 ± 11	n.d.	undefined
[³H]OXD-2314	3.6 ± 0.7	2.4 ± 0.6	2.1 ± 0.3	1.1 ± 0.2	62 ± 8	5.1 ± 0.4	undefined
		2.3 ± 0.3*	2.2 ± 0.4*				

Minor issues

Page 11 line 179: left square bracket missing, [³H]PI-2620

Page 14 line 251: delete "PET" in figure legend "...of female and male rat PET brain..."

The spelling errors have been corrected in the updated manuscript

REVIEWER COMMENTS

Reviewer #1 (Remarks to the Author):

The authors made major efforts to address the comments raised by reviewers and the quality and completeness of the manuscript have significantly improved. This is case particularly for the additional PK data in NHP.

Some concerns remain for the autoradiography experiments. Figure S21 show that the specific signal in PSP is in a similar range to HC. The text should be rephrased reflecting the data and a possible explanation of the difference between the binding data (Table 2) and the autoradiography data (Fig. S21) should be provided.

In addition high-resolution autoradiography experiments are only shown for CBD while the text refers to PSP tissue. This point should be clarified and data should be shown for both CBD and PSP, particularly because of the unclear results in PSP by classical autoradiography.

Lastly, the new table reporting Bmax (Table 3) contains different values in the "answers to reviewers" vs the manuscript. As values differ significantly, representative saturation binding curves or other plots from which Kd and Bmax values have been derived should be shown. Also, depending on which data set is the correct one, the statement about the lower Bmax in PiD and PD should be corrected.

Reviewer #2 (Remarks to the Author):

The authors have satisfactorily addressed all of my concerns in this revised version of the manuscript. The results of this study are significant, highly impactful and will likely make a difference to the field of 4R tauopathies. I do not have any additional concerns or comments.

Reviewer #3 (Remarks to the Author):

The authors have made significant improvements to the manuscript by incorporating revisions, expanding certain sections of the discussion, and addressing reviewers' requests with the addition of new data. Many of my concerns have been effectively addressed. However, I would like to provide further clarification on one of my previous comments regarding the utilization, discussion, and comparison of scores derived from the in silico prediction models.

It is important to note that the BBB score, CNS MPO, and CNS PET MPO do not predict identical outcomes. While it is acceptable to compare them, it's crucial to acknowledge in the discussion that they assess slightly different aspects. The BBB score predicts passive brain permeability, whereas the other two models evaluate the likelihood of overall success as a CNS drug/CNS PET tracer, while (in addition to

BBB permeability) also considering ADME(T) properties such as non-specific binding and p-gp liability. Therefore, I would suggest the authors reconsider certain phrasings in their text.

For instance, on page 7, lines 126-127, where it states, "CNS MPO and CNS PET MPO also predicted that reducing the lipophilicity would increase the BBB permeability," I recommend rewriting it to, "...would increase the likelihood of a successful CNS drug/PET tracer, respectively."

Similarly, in the discussion section (page 21, lines 370-375), where it mentions, "In silico calculations using the BBB score model indicated that OXD-2314 was likely to be more BBB permeable than OXD-2115, whereas the CNS MPO and CNS PET MPO scores suggested OXD-2314 and OXD-2115 would be equally brain permeable," it could be adjusted to reflect the distinct predictions of the models.

Additionally, I respectfully disagree that it is appropriate to cite these models and then assert that "the values are all within the range for potentially CNS active compounds in each model" (page 7, lines 121-123). I suggest rephrasing to convey that while a candidate may slightly fall below the threshold in all models, the values are not absolute (as demonstrated by the authors). Furthermore, it's important to mention that the authors did not use the same software as the models were trained with, and they also incorporated experimental values for the input of logD7.4. This should be acknowledged when discussing the outcomes derived from these prediction models, as it influences the scoring and prediction.

Additional comments:

Page 9: I recommend to use the compound numbers in the description of the synthetic procedures, and not writing the full IUPAC name. Unless it is a commercially available building blocks/reagents, which normally do not have a compound number in the scheme.

Page 12: The in vitro binding properties of [3H]OXD-2314 to AD, PSP, CBD, PiD and PD tissue homogenates are now nicely complemented with Bmax value for each tissue. However, please check the used unit. To my knowledge, Bmax is most commonly reported as (f)mol/mg tissue homogenate not as nM.

Page 16 lines 282-285: The end of this paragraph "...cerebral blood volume in rats is about 4% of total brain volume." I kindly request the authors to cite appropriate reference when making such a statement.

Typos:

Page 9 line 144: Delete "a" in the following sentence: "OXD-2314 was designed and synthesized in a 5-steps starting..."

Page 11 line 168: Remove the underline (or it is from tack changes) after the plus and minus sign "62 ± 8 nM)"

Page 12 line 194: Aβ

Page 12 line 196: Aβ

Page 14 line 242: 5-Lipoxygenase

Page 25 line 467: "In vitro" should be in italic

Page 29 line 557: acquired

Reviewer #1 (Remarks to the Author):

The authors made major efforts to address the comments raised by reviewers and the quality and completeness of the manuscript have significantly improved. This is case particularly for the additional PK data in NHP.

We thank the Reviewer for appreciating our extensive efforts to significantly improve our manuscript, including PK studies that were carried out in two species and two sexes of non-human primates.

Some concerns remain for the autoradiography experiments. Figure S21 show that the specific signal in PSP is in a similar range to HC. The text should be rephrased reflecting the data and a possible explanation of the difference between the binding data (Table 2) and the autoradiography data (Fig. S21) should be provided.

Figure S21 shows autoradiography in BA28/HP tissue (based on our tissue availability). The expected tau deposition was observed in hippocampal tissue for AD cases; specific binding and tau deposits in aged HC and PSP tissues are not expected to be high in these tissues and aligns with the IHC staining for phospho-tau. Please note that Figure S22 demonstrates good specific binding of the tracer in CBD and PSP cortical tissues performed at another site. PSP tau deposition/specific binding would be expected to be greatest in brainstem or subcortical tissues (i.e./globus pallidus and putamen – see Table 3). These tissues for autoradiography are not presently available to us. In view of the Reviewer's comments, the text has been edited to clarify the above-mentioned points, as follows:

We have deleted the text that stated: "...and elevated radiotracer signal was observed in AD and PSP hippocampal tissues when compared to control."

And replaced/edited the remaining text referring to the aforementioned work as follows:

"Low specific binding was observed in aged healthy control tissues and diseased tissues with lower expected tau distribution, whereas elevated radiotracer signal and good specific binding was observed in tau-positive tissues such as hippocampal AD and cortical CBD and PSP tissues, as confirmed by IHC staining for phospho-tau. High resolution autoradiography with [³H]OXD-2314 using nuclear emulsion aligned with staining for phospho-tau deposits demonstrated by IHC in cortical CBD tissue (**Figures S21-S23 and Table S5**)."

In addition high-resolution autoradiography experiments are only shown for CBD while the text refers to PSP tissue. This point should be clarified and data should be shown for both CBD and PSP, particularly because of the unclear results in PSP by classical autoradiography.

We thank the Reviewer for catching this error. High-resolution autoradiography was performed only to demonstrate that autoradiographic signal aligns with phospho-tau deposition and was not performed across tauopathies. The text is now corrected to read that this high-resolution autoradiography was performed with CBD tissue. This is clarified as stated above: "High resolution autoradiography with [³H]OXD-2314 using nuclear emulsion aligned with staining for phospho-tau deposits demonstrated by IHC in cortical CBD tissue (**Figures S21-S23 and Table S5**)."

Lastly, the new table reporting Bmax (Table 3) contains different values in the "answers to reviewers" vs the manuscript. As values differ significantly, representative saturation binding curves or other plots from which Kd and Bmax values have been derived should be shown. Also, depending on which data set is the correct one, the statement about the lower Bmax in PiD and PD should be corrected.

We thank the Reviewer for pointing this discrepancy. This difference in reported Bmax values arose during a 2nd round of minor revisions to the manuscript after a post-submission enquiry from the Editor, and unfortunately did not get appropriately updated in the "answers to reviewers" document. The Bmax values in the manuscript are the correct data set. The text has been further edited to reflect the correct Bmax values in the frontal cortex and subcortical tissues as follows:

"The B_{max} values of [³H]OXD-2314 in AD, PSP, CBD, PiD, and PD frontal cortex tissue were measured (Table 3) and resulted in the highest B_{max} value in AD tissue at 1990 nM, while PSP, CBD, and PiD frontal cortex B_{max} values ranged from 1390 to 780 nM. PD frontal cortex provided a lower B_{max} value of 240 nM. Subcortical (putamen and globus pallidus) PSP and CBD B_{max} values were higher than cortical values at 2300 and 1600 nM, respectively, while the K_d values of cortical and subcortical PSP and CBD tissues were nearly identical (Table 2)."

Representative homologous binding assay results of [³H]OXD-2314 in AD, PiD, PSP, CBD, and control brain tissues are now included in the Supporting Information, as requested and plots are shown below:

Representative homologous binding assay results of [³H]OXD-2314

Figure S24. Representative homologous binding assay results of [³H]OXD-2314 in AD, PiD, PSP, CBD, and control brain tissues. Two-site binding (high and low affinity) was identified in all brain tissues except control.

Reviewer #2 (Remarks to the Author):

The authors have satisfactorily addressed all of my concerns in this revised version of the manuscript. The results of this study are significant, highly impactful and will likely make a difference to the field of 4R tauopathies. I do not have any additional concerns or comments.

The Authors thank the Reviewer for their support for our manuscript for publication.

Reviewer #3 (Remarks to the Author):

The authors have made significant improvements to the manuscript by incorporating revisions, expanding certain sections of the discussion, and addressing reviewers' requests with the addition of new data. Many of my concerns have been effectively addressed.

We thank the Reviewer for appreciating our extensive efforts to significantly improve our manuscript, particularly with the inclusion of additional experiments and new data.

However, I would like to provide further clarification on one of my previous comments regarding the utilization, discussion, and comparison of scores derived from the in silico prediction models. It is important to note that the BBB score, CNS MPO, and CNS PET MPO do not predict identical outcomes. While it is acceptable to compare them, it's crucial to acknowledge in the discussion that they assess slightly different aspects. The BBB score predicts passive brain permeability, whereas the other two models evaluate the likelihood of overall success as a CNS drug/CNS PET tracer, while (in addition to BBB permeability) also considering ADME(T) properties such as non-specific binding and p-gp liability. Therefore, I would suggest the authors reconsider certain phrasings in their text.

The Reviewer's comment is appreciated and has been addressed point-by-point to the queries below:

For instance, on page 7, lines 126-127, where it states, "CNS MPO and CNS PET MPO also predicted that reducing the lipophilicity would increase the BBB permeability," I recommend rewriting it to, "...would increase the likelihood of a successful CNS drug/PET tracer, respectively."

The Reviewers suggestion has been included in the revised manuscript as follows:

"CNS MPO and CNS PET MPO also predicted that reducing the lipophilicity would increase the likelihood of a successful CNS drug/PET radiotracer, respectively."

Similarly, in the discussion section (page 21, lines 370-375), where it mentions, "In silico calculations using the BBB score model indicated that OXD-2314 was likely to be more BBB permeable than OXD-2115, whereas the CNS MPO and CNS PET MPO scores suggested OXD-2314 and OXD-2115 would be equally brain permeable," it could be adjusted to reflect the distinct predictions of the models.

The Reviewer's comment as been included in the revised manuscript as below:

"*In silico* calculations using the BBB score model indicated that OXD-2314 was likely to be more BBB permeable than OXD-2115. Whereas the CNS MPO and CNS PET MPO scores suggested OXD-2314 and OXD-2115 would be equally suited as CNS drugs/PET radiotracers."

Additionally, I respectfully disagree that it is appropriate to cite these models and then assert that "the values are all within the range for potentially CNS active compounds in each model" (page 7, lines 121-123). I suggest rephrasing to convey that while a candidate may slightly fall below the threshold in all models, the values are not absolute (as demonstrated by the authors). Furthermore, it's important to mention that the authors did not use the same software as the models were trained with, and they also incorporated experimental values for the input of logD_{7.4}. This should be acknowledged when discussing the outcomes derived from these prediction models, as it influences the scoring and prediction.

We thank the Reviewer for these insightful comments about the models. The paragraph below has been added to the manuscript for clarification:

"It is noteworthy that the values acquired when using these models can differ depending on the software or experimental methods used to acquire the physicochemical values and did not use the same software as the models were trained with. In this study, ChemOffice™ was used for all physicochemical properties except logD_{7.4}, which was measured experimentally."

Additional comments:

Page 9: I recommend to use the compound numbers in the description of the synthetic procedures, and not writing the full IUPAC name. Unless it is a commercially available building blocks/reagents, which normally do not have a compound number in the scheme.

While we appreciate the Reviewer's suggestion, we prefer to disclose the full chemical names at first mention in text with the number referencing them in parentheses. We believe that this is an established way of presenting chemical reaction schemes in chemistry literature, and makes it easy to follow and/or

replicate the syntheses.

Page 12: The in vitro binding properties of [3H]OXD-2314 to AD, PSP, CBD, PiD and PD tissue homogenates are now nicely complemented with B_{max} value for each tissue. However, please check the used unit. To my knowledge, B_{max} is most commonly reported as (f)mol/mg tissue homogenate not as nM.

We confirm that nM is the common unit herein for B_{max}. While f(mol)/mg of wet tissue is a commonly used unit for B_{max} it needs to be converted when considering B_{max}/K_d ratios, etc. which are therefore both presented in the same units (nM), and facilitates our analysis and comparison of the radioligand binding data.

Page 16 lines 282-285: The end of this paragraph "...cerebral blood volume in rats is about 4% of total brain volume." I kindly request the authors to cite appropriate reference when making such a statement.

We have added the following citation and reworded to state that "...cerebral blood volume in rats is approximately 4-5% of total brain volume."^{ref}

^{ref}Sandor P, Cox-van Put J, de Jong W, de Wied D. Continuous measurement of cerebral blood volume in rats with the photoelectric technique: effect of morphine and naloxone. Life Sci. 1986 Nov 3;39(18):1657-65. doi: 10.1016/0024-3205(86)90163-3.

Typos:

Page 9 line144: Delete "a" in the following sentence: "OXD-2314 was designed and synthesized in a 5-steps starting..."

Page 11 line 168: Remove the underline (or it is from tack changes) after the plus and minus sign "62 ± 8 nM)"

Page 12 line 194: Aβ

Page 12 line 196: Aβ

Page 14 line 242: 5-Lipoxygenase

Page 25 line 467: "In vitro" should be in italic

Page 29 line 557: acquired

We thank the Reviewer for catching these typos, which are all corrected in the revised manuscript, except 5-Lipoxygenase which we confirmed is accurately stated as 5-Lipoxygenase.

REVIEWER COMMENTS

Reviewer #1 (Remarks to the Author):

I would like to thank the authors for the additional information and explanations provided, particularly for having clarified the mismatch in Table 3 and that the high resolution autoradiography was only performed on CBD tissue.

Concerning the autoradiography data on PSP tissue, in my view the data set is still not complete. In one experiment (Fig. S21) there is no clear difference between the specific signal in PSP vs the HV tissue, despite the presence of some AT8 staining (particularly in the PSP case 1). In the second experiment (Fig. S22) there is some specific signal in a cortical tissue from a PSP donor however no control tissue are shown for this area nor AT8 staining to prove the presence of pathological Tau. If no additional data can be added in Figure S22 (control tissue and/or AT8 IHC), I would suggest to modify the paragraph below describing the autoradiography results only in AD and CBD.

“Low specific binding was observed in aged healthy control tissues and diseased tissues with lower expected tau distribution, whereas elevated radiotracer signal and good specific binding was observed in tau-positive tissues such as hippocampal AD and cortical CBD, as confirmed by IHC staining for phospho-tau. High resolution autoradiography with [3H]OXD-2314 using nuclear emulsion aligned with staining for phospho-tau deposits demonstrated by IHC in cortical CBD tissue (Figures S21-S23 and Table S5).”

A separate paragraph should be added to explain the less clear results with PSP tissue. In addition the figure legend of Figure S22 still need to be updated to reflect the new conclusions reported in the text (no difference between PSP and control tissue).

Reviewer #3 (Remarks to the Author):

The authors have satisfactorily addressed all of my concerns in this version of the manuscript. I do not have any further concerns or comments.

REVIEWER COMMENTS

Reviewer #1 (Remarks to the Author):

I would like to thank the authors for the additional information and explanations provided, particularly for having clarified the mismatch in Table 3 and that the high resolution autoradiography was only performed on CBD tissue.

Concerning the autoradiography data on PSP tissue, in my view the data set is still not complete. In one experiment (Fig. S21) there is no clear difference between the specific signal in PSP vs the HV tissue, despite the presence of some AT8 staining (particularly in the PSP case 1). In the second experiment (Fig. S22) there is some specific signal in a cortical tissue from a PSP donor however no control tissue are shown for this area nor AT8 staining to prove the presence of pathological Tau. If no additional data can be added in Figure S22 (control tissue and/or AT8 IHC), I would suggest to modify the paragraph below describing the autoradiography results only in AD and CBD.

“Low specific binding was observed in aged healthy control tissues and diseased tissues with lower expected tau distribution, whereas elevated radiotracer signal and good specific binding was observed in tau-positive tissues such as hippocampal AD and cortical CBD, as confirmed by IHC staining for phospho-tau. High resolution autoradiography with [³H]OXD-2314 using nuclear emulsion aligned with staining for phospho-tau deposits demonstrated by IHC in cortical CBD tissue (Figures S21-S23 and Table S5).”

A separate paragraph should be added to explain the less clear results with PSP tissue. In addition the figure legend of Figure S22 still need to be updated to reflect the new conclusions reported in the text (no difference between PSP and control tissue).

The above-mentioned paragraph has been corrected reworded to clarify the data and explain the less clear results with PSP tissue, as follows:

“Low specific binding was observed in aged healthy control and hippocampal PSP tissues, whereas elevated radiotracer signal and good specific binding was observed in hippocampal AD as well as cortical PSP and CBD tissues. Radiotracer signals in HC, AD and CBD tissues aligned with IHC staining for phospho-tau. High resolution autoradiography with [³H]OXD-2314 using nuclear emulsion aligned with staining for phospho-tau deposits demonstrated by IHC in cortical CBD tissue (Figures S21-S23 and Table S5).”

And, the respective figure legends have been updated as follows:

Figure S21. [³H]OXD-2314 binding in HC, AD and PSP hippocampal tissue. Total [³H]OXD-2314 (3 nM) binding is shown along with displacement by unlabeled OXD-2314 (10 μM) compared to AT8 immunostaining for tau. Low specific binding was observed in aged healthy control tissues and PSP tissues with low tau distribution, whereas elevated radiotracer signal and good specific binding was observed in hippocampal AD, as confirmed by IHC staining for phospho-tau.

Figure S22. [³H]OXD-2314 binding in CBD and PSP cortical tissue. Total [³H]OXD-2314 (3 nM) binding is shown along with displacement by unlabeled OXD-2314 (3 μM) and AT8 immunostaining for tau in CBD tissue.

Reviewer #3 (Remarks to the Author):

The authors have satisfactorily addressed all of my concerns in this version of the manuscript. I do not have any further concerns or comments.

We thank Reviewer 3 for their support of our manuscript for publication.